# Machine learning-based Alpine treeline ecotone detection on Xue Mountain in Taiwan

Geng-Gui Wang[1], Min-Chun Liao[2], Wei Wang[3,] Hui Ping Tsai[1,4,*], Hsy-Yu Tzeng[5]

5  [1]Department of Civil Engineering, and Innovation and Development Center of Sustainable Agriculture, National Chung Hsing University, Taichung City 402, Taiwan (R.O.C.)
[2] Chiayi Research Center, Taiwan Forestry Research Institute, Chiayi City 600, Taiwan (R.O.C.)
[3] Experimental Forest, National Chung Hsing University, No. 145 Xingda Rd., Taichung City 402, Taiwan (R.O.C.)
[4] i-Center for Advanced Science and Technology, National Chung Hsing University, Taichung City 402, Taiwan (R.O.C.)
10  [5] Department of Forestry, National Chung Hsing University, Taichung City 402, Taiwan (R.O.C.)

*Correspondence to*: Hui-Ping Tsai (email)

**Abstract.** Taiwan is characterized by high mountains density, with over 200 peaks exceeding 3,000 meters in elevation. The alpine treeline ecotone (ATE) is a transitional zone between different vegetation types. The species distribution, range 15  variations, and movement patterns of vegetation within the ATE are crucial indicators for assessing the impact of climate change and warming on alpine ecosystems. Therefore, this study focuses on the Xue Mountain glacial cirques in Taiwan (approximately 400 hectares) and utilizes WorldView-2 satellite images from 2012 and 2021 to compute various vegetation indices and texture features (GLCM). By integrating these features with Random Forest (RF) and U-Net models, we developed a classification map of the ATE in Xue Mountain. We analyzed changes in bare land, forest, krummholz, and shadows within 20  the ATE from 2012 to 2021. The results indicate that the classification accuracy reached an overall accuracy (OA) of 0.838 when incorporating raw spectral bands along with vegetation indices and texture features (GLCM) (77 features in total). Feature importance ranking and selection reduced training time by 14.3% while ensuring alignment between field survey treeline positions and classification results. From 2012 to 2021, tree cover density increased, with the total forest area expanding by approximately 10.09 hectares. The upper limit of forest distribution shifted upslope by $32.00 \pm 4.00$ m, with the 25  most significant area changes occurring between 3,500 and 3,600 m, while the 3,700 to 3,800 m range remained relatively stable. This study integrates remote sensing imagery with deep learning classification methods to establish a large-scale ATE classification map. The findings provide a valuable reference for the sustainable management of alpine ecosystems in the Xue Mountain glacial cirques in Taiwan.

## 1 Introduction

Taiwan is located in the subtropical region of Southeast Asia, with elevations ranging from nearly 4,000 m, fostering diverse ecosystems types and rich biodiversity (Lin et al., 2021). The island contains more than 200 mountains exceeding 3,000 meters in elevation (Kuo et al., 2022), making it one of the highest-density alpine islands in the world (Chen, 2017). Alpine zone ecosystems are particularly vulnerable to environmental change due to their high environmental heterogeneity 35  and limited species migration distances, especially when compared to broader latitudinal climate gradients and more resilient lowland regions (Engler et al., 2011; Huss et al., 2017; Li et al., 2018; Zheng et al., 2020). The transition zone between trees and treeless vegetation in alpine ecosystems is known as the alpine treeline or the alpine treeline ecotone  (ATE) (Körner 2012). The ecological processes and changes in this zone are considered indicators of climate change (Chen et al., 2022), reflecting the interactions of climate, topography, species composition, and disturbance history (Loranger et al., 2016; Johnson et al., 40  2017; Mohapatra et al., 2019; Bader et al., 2021). Based on many studies, changes in the ATE illustrate the impacts of climate change on mountain ecosystems, such as the upward migration of tree species and increased tree density. However, these shifts

are also influenced by other drivers, including land-use history, altered disturbance regimes (e.g., fire, landslide, windthrows), herbivory pressure, and species-specific physiological traits. Moreover, cascading effects among these disturbances can further amplify ecological responses and accelerate treeline dynamics (Wang et al., 2016; Johnson et al., 2017; Du et al., 2018; Mohapatra et al., 2019, Stritih et al., 2024; Lu et al., 2025).

Machine learning is increasingly being combined with high-resolution remote sensing to enhance land-cover and forest-type classification. Among the numerous algorithms, each model has its own strengths. Random forests (RF) and support vector machines (SVM) have gained widespread use due to their robustness and effectiveness in processing multispectral data with limited training samples (Belgiu and Drăguţ, 2016; Jombo et al., 2020; Jackson and Adam, 2021). RF, in particular, exhibits strong interpretability and stability in heterogeneous environments. In contrast, deep learning models such as U-Net demonstrate superior ability to capture both spectral and spatial information, achieving high segmentation accuracy in complex landscapes (Ronneberger et al., 2015; Freudenberg et al., 2019; Wagner et al., 2019). Recent comparative studies further demonstrate that RF and SVM remain reliable and interpretable choices for multispectral classification when training data is limited or imbalanced. At the same time, U-Net and other convolutional neural network (CNN) architectures generally provide superior spatial accuracy and boundary delineation in high-resolution or well-labeled datasets. Furthermore, transferability analysis shows that U-Net models generally have better generalization capabilities in large or heterogeneous regions, while RFs tend to perform more consistently in small sample or sparsely labeled scenarios (Boston et al., 2022; Ge et al., 2021; Nigar et al., 2024).

In Taiwan, many alpine forest studies have been conducted through field surveys using an ecological approach at relatively small spatial scales, focusing on flowering phenology and growth assessment for specific tree species (Chiu et al., 2022; Liao et al., 2023a; Kudo et al., 2024). In recent years, Chung et al. (2021) used Landsat 8 imagery combined with support vector machine (SVM) classification to examine timberline dynamics on Taiwan's highest peak, Yushan, revealing the influence of temperature on timberline shifts. The Xue Mountain, Taiwan's second-highest peak, has also been the subject of long-term ecological monitoring (Chung et al., 2021; Liao et al., 2023b). However, extensive targeting alpine treeline ecotone (ATE) dynamics remains lacking. This study provides the first comprehensive analysis of changes in the ATE landscape in Taiwan's Xue Mountain glacial cirque region. It uses ultra-high-resolution WorldView-2 satellite imagery with Random Forest (RF) and U-Net models. The aim is to quantify spatiotemporal changes between 2012 and 2021.

## 2 Materials and methods

### 2.1 Study site

The Xue Mountain glacial cirques are located in Shei-Pa National Park in north-central Taiwan, covering an area of approximately 400 hectares (Fig. 1). The central peak of Xueshan has an elevation of 3,886 m. The cirque serves as a crucial habitat for Taiwan's endemic species, the Yushan Juniper (*Juniperus morrisonicola*), Yushan rhododendron (*Rhododendron pseudochrystam*), and the Taiwan fir (*Abies kawakamii*), which is primarily distributed at elevations between 3,000 and 3,600 m. Most ecological studies conducted in this research area have focused on Taiwan fir forests, with several researchers estimating wood volumes, competitive pressure, forest structure, and spatial distribution of the species primarily through field surveys conducted below the alpine treeline ecotone (Li et al., 2021; Wang et al., 2021; Chiu et al., 2022; Liao et al., 2023a; Liao et al., 2023b). In contrast, relatively little attention has been given to the dynamics of treeline ecotone shifts.

In this study, we define the treeline ecotone not as a fixed linear boundary but as a transitional zone where krummholz, such as Yushan Juniper and Yushan rhododendron, begin to appear within the alpine talus slope (Liao. 2016; Liao et al., 2023a). This ecotone represents an area of ecological transition from subalpine forest to alpine vegetation.

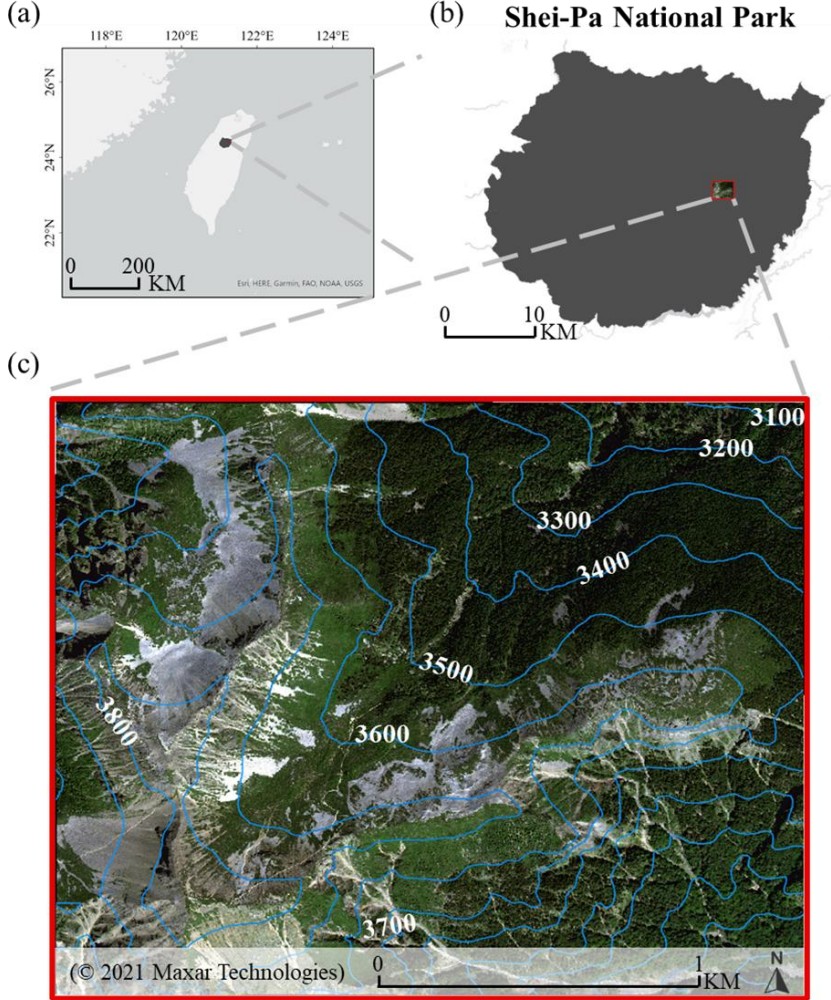

**Figure 1. Study area. (a) Geographic location of Shei-Pa National Park in north-central Taiwan. (b) Treeline ecotone study area located in the Xue Mountain glacial cirques within Shei-Pa National Park. (c) WorldView-2 image showing the research area with topographic contours.**

## 2.2 Research flow

This study utilized WorldView-2 satellite imagery from 2021 to extract raw spectral bands, vegetation indices, and texture features. Starting with the eight spectral bands, vegetation indices, and texture features were sequentially added to form four different feature combinations. Classification models were developed using the RF and U-Net models, and the optimal model is selected. This model is then applied to 2012 imagery to map the distribution of the alpine treeline and analyzed changes over the decade. The research workflow was illustrated in Fig. 2.

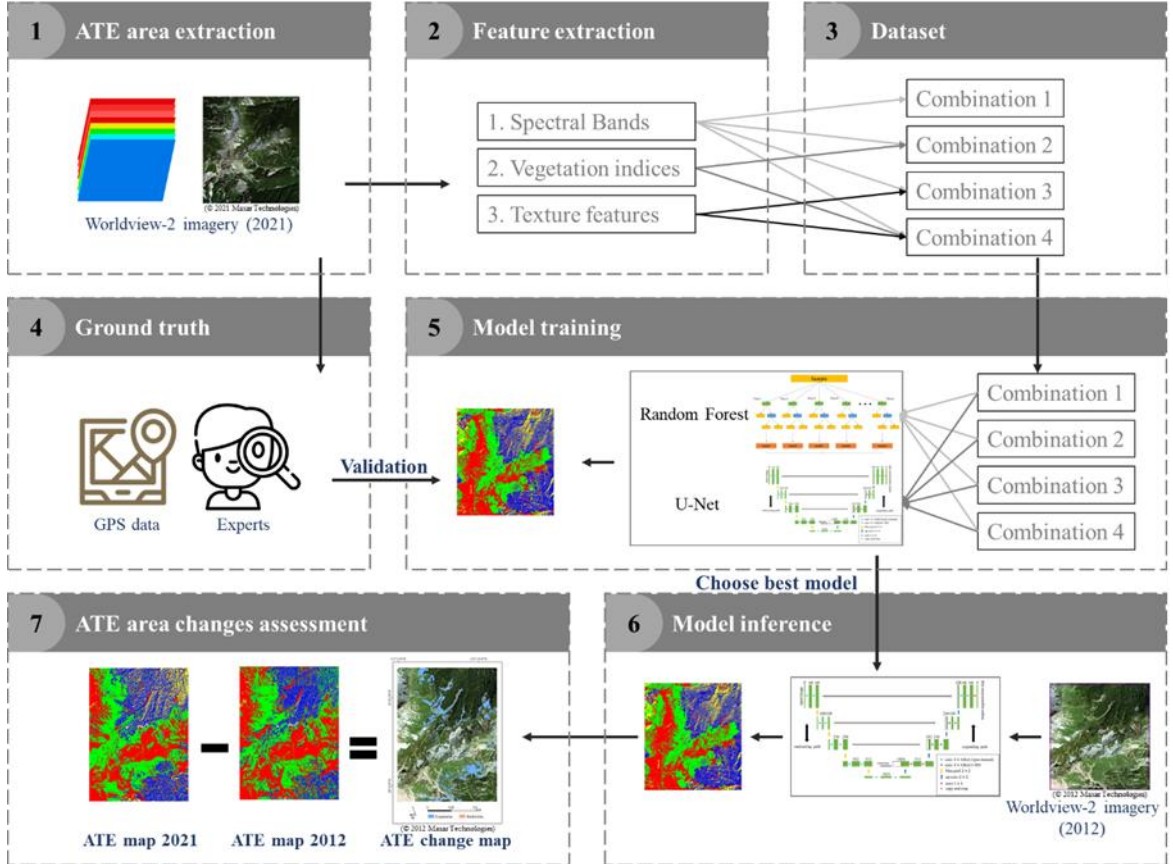

**Figure 2. Research flow for classifying WorldView-2 images of a treeline ecotone on Mt. Xue in Taiwan to detect treeline changes. The process begins with WorldView-2 satellite image acquisition, followed by feature extraction (spectral bands, vegetation indices, and texture features), model training using Random Forest (RF) and U-Net, accuracy evaluation, feature selection, and temporal analysis of alpine treeline changes between 2012 and 2021.**

## 2.3 Research data

The research data sources were categorized into satellite imagery and field surveys, with satellite imagery as the primary source and field surveys used as supplementary validation to ensure the accuracy of the treeline boundary. WorldView-2 was an environmental monitoring satellite operated by Maxar Technologies Inc. (Colorado, USA). It was launched on October 8, 2009, and its geolocation accuracy, is reported to be within 3 meters. Depending on the spatial resolution, the revisit time ranges from 1.1 to 3.7 days.

The satellite provided two imaging modes: panchromatic and multispectral. The spatial resolution was 0.41 m in the panchromatic mode, and the spectral range spans 450–800 nm. This mode offered high spatial resolution, allowing for detailed image representation. In the multispectral mode, the spatial resolution was 1.64 m, and the spectral range extended from 400 to 1040 nm, covering eight spectral bands, as shown in Table 1. To enhance spatial detail, all multispectral bands were pansharpening using the corresponding high-resolution panchromatic band, yielding a uniform spatial resolution of 0.4 meters across all datasets used for feature extraction. The pansharpened multispectral imagery was the basis for deriving vegetation indices and texture features.

Two orthorectified, cloud-free WorldView-2 images acquired on November 3, 2012, and September 26, 2021, were obtained from RiChi Technology Co., Ltd. (New Taipei City, Taiwan). Due to partial cloud coverage in the 2012 imagery, only approximately 150 ha of cloud-free area was used for subsequent temporal comparisons. In contrast, the 2021 imagery covered the entire study region (about 400 ha) (Fig. 3). The 2021 image was therefore used for model training and feature optimization, and both images were used for analysis within the common cloud-free area to ensure comparability. Both images were captured in the autumn season when vegetation had entered its dormant phase, minimizing the influence of phenological variability. Histogram matching was applied to ensure radiometric consistency across the two images. In addition, Global

Navigation Satellite System (GNSS) devices were used to record field survey points in 2023, which were subsequently used to verify alpine treeline ecotone (ATE) positions and assist in manual ground truth labeling.

**Table 1. Spectral characteristics of WorldView-2 satellite bands**

| Band | Spectral range (nm) |
|---|---|
| Costal Blue (CB) | 400-450 |
| Blue (B) | 450-510 |
| Green (G) | 510-580 |
| Yellow (Y) | 585-625 |
| Red (R) | 630-690 |
| Red Edge (RE) | 703-745 |
| Near Infrared 1 (NIR1) | 770-895 |
| Near Infrared 2 (NIR2) | 860-1040 |

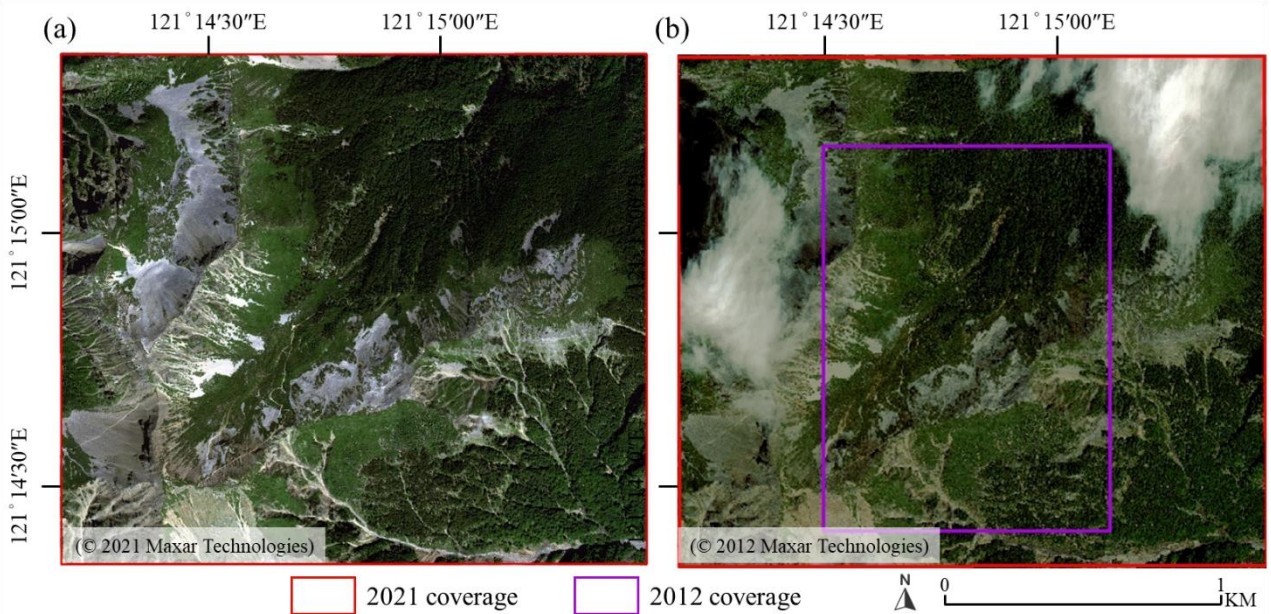

**Figure 3. Extent of the 2012 and 2021 WorldView-2 images in the Xue Mountain glacial cirques. (a) 2021 image and (b) 2012 image. The red polygon shows the full 2021 coverage (400 ha), while the transparent area indicates the 2012 image affected by cloud contamination. The purple outline delineates the cloud-free overlap area (150 ha) used for temporal change analysis.**

### 2.4 Vegetation indices

The reflectance spectrum of plant leaves can reflect their internal physiological status, such as chlorophyll content, water content, intercellular spaces, and cell walls (Croft et al., 2014; Xu et al., 2023; Neuwirthová et al., 2024; Špundová et al., 2024). The frequently discussed spectral bands include red (R), the red edge (RE), and the near-infrared (NIR) bands. Derived vegetation indices, such as the Normalized Difference Vegetation Index (NDVI) and the Enhanced Vegetation Index (EVI), have been widely used (Rouse et al., 1974; Huete et al., 2002). Additionally, some studies have suggested that the blue (B) and green (G) bands can be used to monitor vegetation phenology and forests. For example, indices such as the Green Chromatic Coordinate (GCC) and the Excess Green Index (ExG) have been developed for this purpose (Sonnentag et al., 2012; Larrinaga and Brotons, 2019). Since image acquisition was affected by terrain, leading to shadow occurrences that influence classification accuracy, this study also planned to adopt the Shadow-Eliminated Vegetation Index (SEVI) (Jiang et al., 2019). In this study, 11 vegetation indices were used, as summarized in Table 2.

**Table 2. List of vegetation indices and their formulas derived from spectral bands.**

| Vegetation Index | Formula | Reference |
|---|---|---|
| Difference Vegetation Index (DVI) | $NIR - R$ | Richardson and Wiegand, 1977 |
| Enhanced vegetation index (EVI) | $2.5 \times \dfrac{(NIR - R)}{(NIR + 6 \times R - 7.5 \times B + 1)}$ | Huete et al., 2002 |
| Excess Blue Vegetation Index (ExB) | $\dfrac{1.4 \times B - G}{G + R + B}$ | Mao et al., 2003 |
| Excess Green Index (ExG) | $\dfrac{2 \times G - R - B}{G + R + B}$ | Woebbecke et al., 1995 |
| Excess Green minus Excess Red (ExGR) | $ExG - ExR$ | Meyer and Neto, 2008 |
| Excess Red Vegetation Index (ExR) | $\dfrac{1.4 \times R - G}{G + R + B}$ | Meyer et al., 1999 |
| The Green Chromatic Coordinate (GCC) | $G/(R + G + B)$ | Gillespie et al., 1987 |
| Normalized difference index (NDI) | $\dfrac{G - R}{G + R}$ | Gitelson and Merzlyak, 1994 |
| Normalized difference vegetation index (NDVI) | $\dfrac{NIR - R}{NIR + R}$ | Rouse et al., 1974 |
| Ratio Vegetation Index (RVI) | $\dfrac{NIR}{R}$ | Jordan, 1969 |
| Shadow- Eliminated Vegetation Index (SEVI) | $RVI + f(\Delta) \times \dfrac{1}{R}$ | Jiang et al., 2019 |

**2.5 Texture Feature**

With improvements in satellite image resolution, a single ground object may consist of multiple pixels, making spatial information increasingly important for image interpretation (Wang et al., 2015). Texture features describe the spatial
 arrangement and structural patterns of objects within an image, providing complementary information to spectral reflectance. This allows for better discrimination of land cover types with similar spectral characteristics. Texture analysis methods can be categorized into spectral, statistical, and structural approaches, with the Gray Level Co-occurrence Matrix (GLCM) in statistical approaches being the most commonly used (Hsu, 1978). Following the parameter settings suggested by previous studies (Guo et al., 2020; Sibiya et al., 2021), texture features were extracted to enhance spatial information for classification.
 In this study, a moving window size of $7 \times 7$ was applied based on their findings, which provided an effective balance between detail and noise in texture analysis. Therefore, this study adopted a $7 \times 7$ moving window to compute the GLCM matrix for each of the eight bands, analysing seven statistical metrics, resulting in 56 texture features. The seven statistical metrics used in this study are listed in Table 3.

**Table 3. Description of texture features calculated using the gray-level co-occurrence matrix (GLCM).**

| Texture Feature | Formula | Reference |
|---|---|---|
| Contrast (Con) | $\sum_{i,j=0}^{N-1} P_{i,j}(i - j)^2$ | Yuan et al., 1991 |
| Dissimilarity (Dis) | $\sum_{i,j=0}^{N-1} P_{i,j}|i - j|$ | Rubner et al., 2002 |
| Energy (Ene) | $\sum_{i,j=0}^{N-1} P_{i,j}{}^2$ | Hall-Beyer, 2017 |

| Entropy (Ent) | $\sum_{i,j=0}^{N-1} P_{i,j}\left(-lnP_{i,j}\right)$ | Yuan et al., 1991 |
|---|---|---|
| Homogeneity (Hom) | $\sum_{i,j=0}^{N-1} \dfrac{P_{i,j}}{1+(i-j)^2}$ | Hall-Beyer, 2017 |
| Mean (M) | $\sum_{i,j=0}^{N-1} iP_{i,j}$ | Materka and Strzelecki, 1998 |
| Variance (Var) | $\sum_{i,j=0}^{N-1} P_{i,j}(i-\text{Mean})^2$ | Materka and Strzelecki, 1998 |

$P_{i,j}$ is the gray-level co-occurrence matrix after normalization.

## 2.6 Methods

### 2.6.1 Random Forest (RF)

Random Forests (RF) was an ensemble classifier widely used in remote sensing due to its ability to handle high-dimensional data. It generates multiple decision trees (DTs), where each tree made predictions based on observed features through a series of decision-making steps, ultimately concluding the target variable. Decision trees, also known as classification trees, were a type of predictive model. Random forests used the Bagging algorithm (Bootstrap Aggregating) as their core classification mechanism. The process began by randomly sampling the data to create training datasets. After each sampling, the selected data points were returned to the dataset for the next round of sampling (bootstrap sampling) (Breiman, 2001). This process was repeated multiple times, resulting in several training datasets, which were then used to train multiple decision trees. This approach allowed for scenarios where specific data points were sampled multiple times while others may not. Each decision tree selected a random subset of features at each node to determine the best split, ultimately generating predictions from each tree. The final classification result was determined by aggregating the predictions of all decision trees through a majority voting approach, which means that each tree casts one "vote" for a class label, and the class receiving the most votes becomes the final prediction. To evaluate the importance of each feature, the Random Forest model uses the Gini Index (Breiman, 2001), which measures the impurity of a node. A node represents a point in the tree where the dataset is split based on a feature, with each node divided using the best split among a random subset of explanatory variables (Breiman, 2001). A lower Gini value indicates better class separation. The Gini Index for a node *m* is calculated as follows:

$$Gini_m = \sum_{k=1}^{k} \hat{p}_{mk}(1-\hat{p}_{mk}) , \tag{1}$$

Where $\hat{p}_{mk}$ was the probability of a sample at node *m* belonging to class *k*, and *K* was the total number of classes. The Gini Index also supported the out-of-bag (OOB) error estimation and was commonly used to determine feature importance in classification tasks. Feature importance quantifies how much each variable reduces node impurity and contributes to improving classification accuracy across all trees in the forest (Belgiu and Drăguţ, 2016; Breiman, 2001; Chen et al., 2023).

### 2.6.2 U-Net

Ronneberger et al. (2015) proposed the original U-Net model, which was devolved from the fully convolutional network (FCN) and was initially designed for biomedical image segmentation. The U-Net model consists of a contracting path (downsampling) and an expanding path (upsampling) (Ronneberger et al., 2015). Similar to FCN, U-Net did not use fully connected layers, and its convolutional layers significantly reduced the amount of training data required while allowing inputs of different sizes. Before entering the contracting or expanding path, the data underwent two consecutive convolutional layers,

which helped the network extract target features more effectively. This process also enhanced the integration of fine details with feature maps, thereby improving segmentation quality. Each convolutional layer was followed by a rectified linear unit (ReLU) activation function, which enhances training efficiency without affecting model accuracy. The pooling layer at the bottom served as a nonlinear form of downsampling, reducing the spatial size of the data, decreasing the number of parameters and computational costs, and helping to control overfitting. Since U-Net lacked fully connected layers, it effectively minimized information loss caused by downsampling while preserving finer image details.

### 2.6.3 Data set

The WorldView-2 satellite imagery consists of eight spectral bands. Based on these eight bands, this study derived 13 vegetation indices and 56 texture features, resulting in 77 feature variables. The original eight bands were incrementally combined with vegetation indices and texture features, forming four different feature combinations (Table 4).

Ground truth data in the study area were manually labeled using a pixel-based approach and categorized into four classes: (1) Bare land, referring to areas of exposed soil, rock surfaces, or sparsely vegetated ground; (2) Forest, defined as regions with dense, continuous tree canopy cover; (3) Krummholz, representing stunted, shrub-like trees typically found at high elevations near the treeline and shaped by wind or snow pressure (Liao et al., 2023a); and (4) Shadow, representing regions with low reflectance caused by topographic shading or solar angle effects. The class definitions were established based on visual inspection and field knowledge of the study area (Fig. 4). The labeling process was independent and performed by visually interpreting the pansharpened RGB composite imagery, referencing known terrain characteristics, and assisted by field-collected GNSS survey points.

Each image ($5380 \times 4671$ pixels) was segmented into 110 non-overlapping patches of $512 \times 512$ pixels. The dataset split was performed at the patch level, to avoid spatial autocorrelation and data leakage (Roberts et al., 2017). Specifically, 80% of the patches were randomly selected for training and validation (with a 75/25 split), and the remaining 20% were used as an independent test set. In total, 66 patches were used for training, 22 for validation, and 22 for testing.

**Table 4. Definitions of the four feature combinations used in model training. The table shows the input feature types and their corresponding dimensionality.**

| Feature combinations | Input feature | Feature Dimension |
|---|---|---|
| 1 | spectral band | 8 |
| 2 | spectral band, vegetation indices | 21 |
| 3 | spectral band, texture features | 64 |
| 4 | spectral band, vegetation indices, texture features | 77 |

**(a) RGB image**

**(b) Ground truth**

Bare land    Forest    Krummholz    Shadow

**Figure 4. Ground truth label generation for land cover classification. (a) WorldView-2 RGB composite image from 2021; (b) manually annotated labels showing four classes: forest, krummholz, bare land, and shadow.**

### 2.6.4 Evaluation Index

This study uses overall accuracy (OA), F1-score, and the Kappa coefficient as assessment metrics to evaluate classification accuracy. The formulas for each metric are explained below.

$$OA = \frac{TP+TN}{TP+FP+TN+FN}, \tag{2}$$

$$\text{F1} - \text{score} = \frac{2 \times TP}{2 \times TP+FP+FN}, \tag{3}$$

$$\text{Kappa} = \frac{P_o - P_e}{1 - P_e}, \text{ with} \tag{4}$$

$$P_o = \frac{TP+TN}{TP+FP+TN+FN}, \text{ and} \tag{5}$$

$$P_e = \frac{(TP+FN) \times (TP+FP)+(FP+TN) \times (FN+TN)}{(TP+FP+TN+FN)^2}, \tag{6}$$

Among them, *TP* (true positive), *TN* (true negative), *FP* (false positive), and *FN* (false negative).

### 2.6.5 Bootstrapping

The bootstrap resampling method was a nonparametric approach used to estimate the variability and confidence intervals (CIs) of a statistic by repeatedly resampling with replacement from the original dataset. It enabled robust inference without assuming a specific data distribution (Efron and Tibshirani, 1993). The percentile method was commonly used, in which the 2.5th and 97.5th percentiles of the bootstrap distribution defined the 95% CI (Davison and Hinkley, 1997). To ensure stable and reliable estimates, between 1000 and 10,000 bootstrap iterations were generally recommended (Davison and Hinkley, 1997), with at least 5000 replicates providing sufficient accuracy for most applications (Carpenter and Bithell, 2000).

## 3. Results

### 3.1 Feature combination and feature importance analysis

This study employed Random Forest (RF) and U-Net models with four feature combinations to examine land cover classes in Taiwan's Xue Mountain glacial cirques in the alpine treeline ecotone (ATE) region. Four land cover classes —bare land, forest, krummholz, and shadow —were investigated using feature combinations of spectral bands (8 features), vegetation indices (13 features), and texture features (56 features). The classification results of the RF and U-Net models with four feature

combinations were compared in detail (Fig. 5 and Table 5). In general, the RF model demonstrated stable, robust classification performance across various feature dimensions. Specifically, the average F1-score of the RF model ranged from 0.823 to 0.839, the overall accuracy (OA) ranged from 0.817 to 0.830, and the Kappa coefficient ranged from 0.751 to 0.768 (Table 5). Among all classes, shadow and bare land achieved the highest F1-scores, both exceeding 0.85, while forest and krummholz maintained moderate but stable accuracy, ranging from 0.75 to 0.83. Additionally, the combination 4 yielded the highest F-1 score in forest and krummholz classes, indicating that the RF model improved when vegetation indices and texture features were combined with spectral information.

Furthermore, the U-Net model exhibited a marked improvement after incorporating more features. The F1-score for the forest class increased significantly from 0.609 for feature combination 1 (spectral bands only) to 0.828 for combination 4 (spectral, vegetation indices, and texture features). Likewise, the F1-score for krummholz improved from 0.696 to 0.778. Bare land and shadow also maintained high accuracy above 0.82 across all combinations. The U-Net's overall performance metrics (F1-score of 0.840, OA of 0.838, and Kappa of 0.778 in combination 4) surpassed those of RF, indicating that the U-Net model benefited substantially from integrating spectral, vegetation, and texture information.

Overall, the results showed that incorporating vegetation indices and texture features improved classification performance, particularly for vegetation classes in the U-Net model. Based on the higher F1-score in combination 2 than combination 3, it implied that vegetation indices contributed more than texture features. However, the highest F1-score was obtained in combination 4, indicating a complementary effect from vegetation indices and texture features. Additionally, the consistency between the classified ATE and field-observed forest–krummholz transitions further confirmed the classification's reliability. Overall, both models maintained stable performance across different feature combinations, supporting the robustness of the proposed approach.

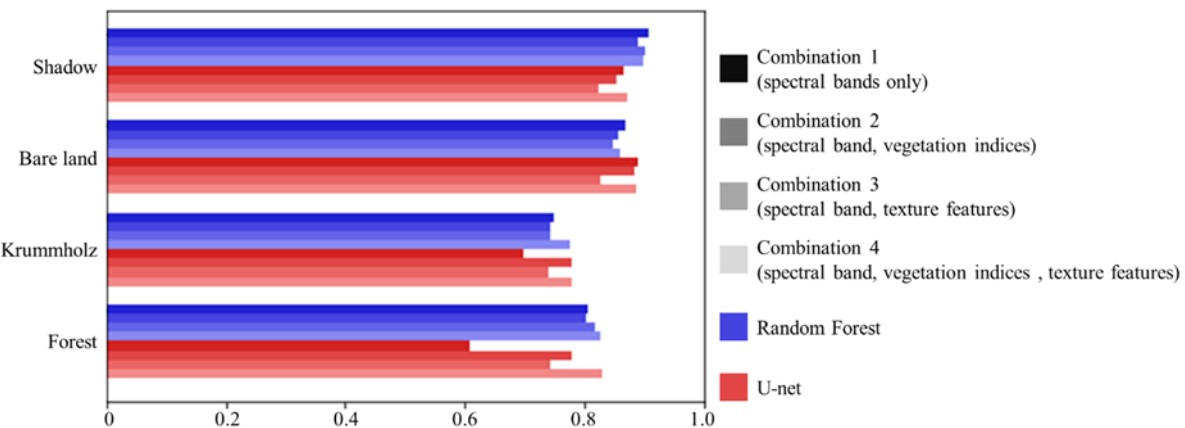

**Figure 5. F1-scores for four land cover classes (forest, krummholz, bare land, shadow) using RF and U-Net models with different feature combinations.**

**Table 5. Evaluation of classification accuracy using different feature combinations and models. Average F1-score, Overall accuracy (OA) and Kappa coefficient are shown for Random Forest (RF) and U-Net models. Numbers in parentheses indicate the number of input features. Bold values indicate the best results for each metric.**

| Feature (DIMs) | Combinations 1(8) | | Combinations 2(21) | | Combinations 3(64) | | Combinations 4(77) | |
|---|---|---|---|---|---|---|---|---|
| Method | RF | U-Net | RF | U-Net | RF | U-Net | RF | U-Net |
| Average F1-score | 0.831 | 0.765 | 0.823 | 0.823 | 0.826 | 0.782 | 0.839 | **0.840** |
| OA | 0.819 | 0.753 | 0.817 | 0.780 | 0.812 | 0.819 | 0.830 | **0.838** |
| Kappa | 0.753 | 0.666 | 0.751 | 0.703 | 0.743 | 0.755 | 0.768 | **0.778** |

Based on the RF and U-Net model results, a further feature importance analysis was conducted to assess individual features in combination 4, comprising 77 features, including spectral bands, vegetation indices, and texture features. The

feature importance analysis results revealed that the cumulative contribution achieved 95% interpretability with 61 features. Additionally, the OA and Kappa values improved slightly to 0.842 and 0.784, respectively. Moreover, computation time was

reduced by 14.3% due to fewer features (Table 6). According to the feature ranking results, spectral bands and vegetation indices ranked higher than texture features, with SEVI, Y, B, G, and NDVI2 identified as the top five features (Fig. 6).

**Table 6. Comparison of model performance before and after feature selection. Training time is presented in hours. The results show reduced training time and slightly improved classification accuracy after feature selection.**

|  | Without feature selection | With feature selection | Difference (%) |
|---|---|---|---|
| Training time(hr) | 7.708 | 6.608 | -14.3 |
| OA | 0.838 | 0.842 | +0.4 |
| Kappa | 0.778 | 0.784 | +0.4 |

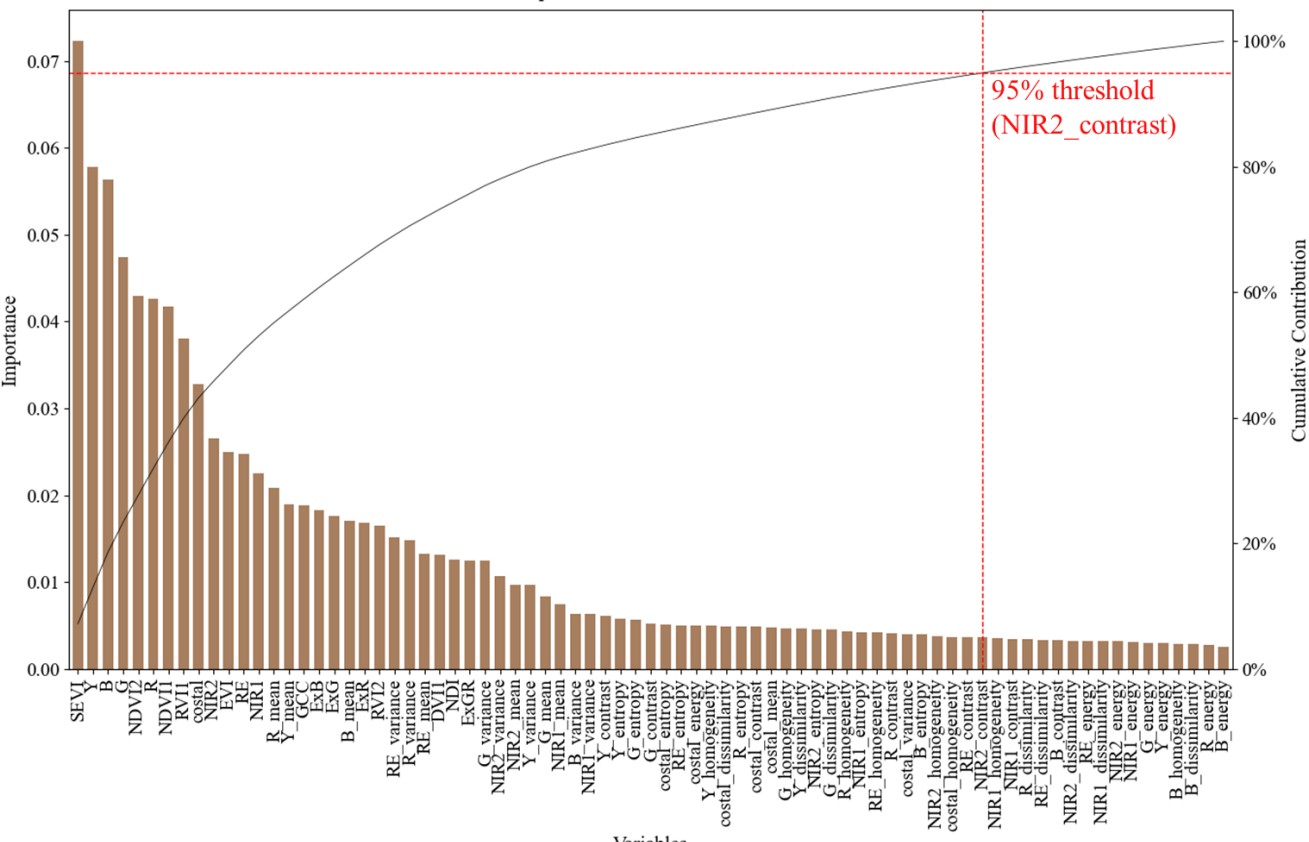

**Figure 6. Feature importance ranking derived from the Random Forest model. Features are ranked based on their contribution to classification accuracy, with the top-ranked features including SEVI, Y (yellow), B (blue), G (green), and NDVI2. Most of the top features are spectral bands and vegetation indices, while texture features rank lower.**

**3.2 Decadal changes in the alpine treeline ecotone (ATE)**

The U-Net model was trained using the 2021 imagery (covering ~400 ha) and applied to classify both the 2012 and 2021 datasets. However, since the 2012 image was affected by cloud cover, only the 150 ha of overlapping cloud-free area was used for the decadal change analysis. The classification results were validated against field survey data collected in 2021, which recorded vegetation types and tree positions for two tracks in the study area. As shown in Fig. 7, the classification results

closely align well with the GNSS-measured tree coordinates recorded during the 2023 field survey. Over the decade, the proportion of forest area increased by 3.4%, indicating a possible trend of green coverage expansion associate with tree growth, denser canopy, or growing saplings. Meanwhile, the proportion of shadow area also increased by 8.5%. which may associate with possible tree growth. Additionally, krummholz and bare land areas decreased by 3.2% and 8.7%, respectively (Table 7).

For the forest category, the forest area expanded by 10.49 hectares and was reduced by 0.4 hectares between 2012 and 2021
(Fig. 8 and Table 8).

Based on the 95th percentile of DEM elevation values for all pixels classified as forest (Fig. 9), the elevation difference increased by 32.00 meters between 2012 and 2021. The 95% confidence interval (± 4.00 meters) was estimated using a bootstrap resampling method (5,000 iterations). Differences in area changes across various elevation ranges are detailed in Table 8, with the most significant changes occurring in the 3,600- to 3,700-m range, which corresponds to the primary treeline ecotone change zone in the Xue Mountain region. In comparison, the most stable area was observed in the 3,700 to 3,800 m range, where minimal forest presence was detected in both 2012 and 2021, reflecting physiological limits of trees.

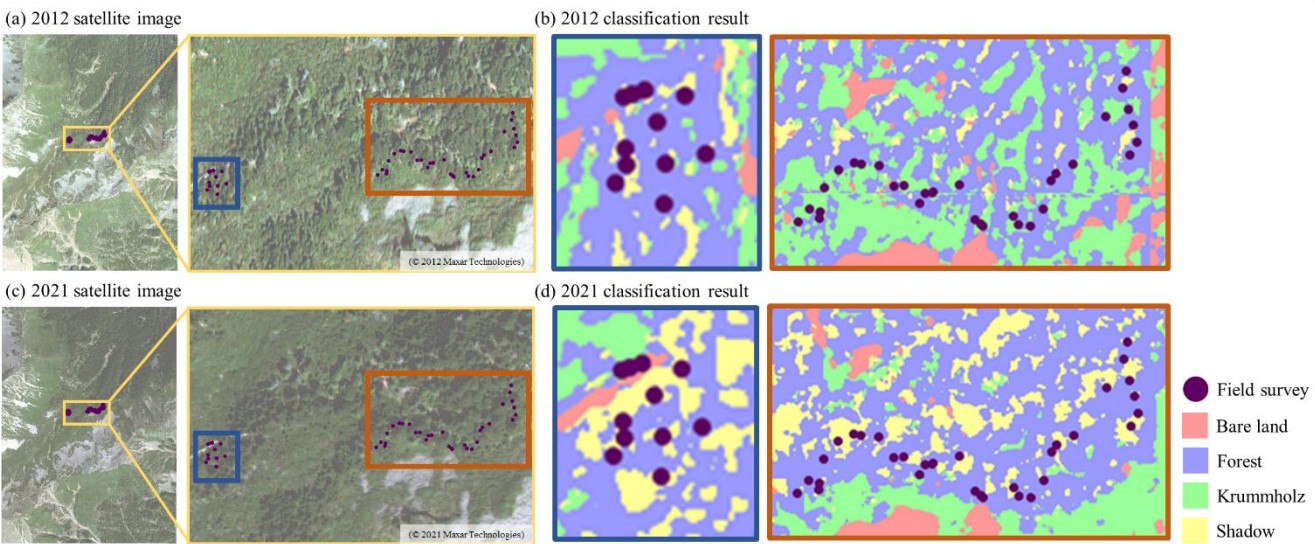

**Figure 7 Comparison of satellite imagery and classification results from 2012 and 2021. Panels (a) and (c) show high-resolution satellite images for 2012 and 2021, respectively. Colored boxes in these images indicate the enlarged areas shown in (b) and (d). Panels (b) and (d) present the classification results of the corresponding enlarged regions using a U-Net model trained with 61 selected features. Triangles mark field survey locations.**

**Table 7. Percentage of each land cover class in 2012 and 2021 classification results. Forest and shadow areas increased over time, while krummholz and bare land decreased.**

| Classification percentage (%) | Year | | Increment / Decrement |
|---|---|---|---|
| | 2012 | 2021 | |
| Forest | 22.5 | 25.9 | +3.4 |
| Krummholz | 36.4 | 33.2 | -3.2 |
| Bare land | 38.1 | 29.4 | -8.7 |
| Shadow | 3.0 | 11.5 | +8.5 |

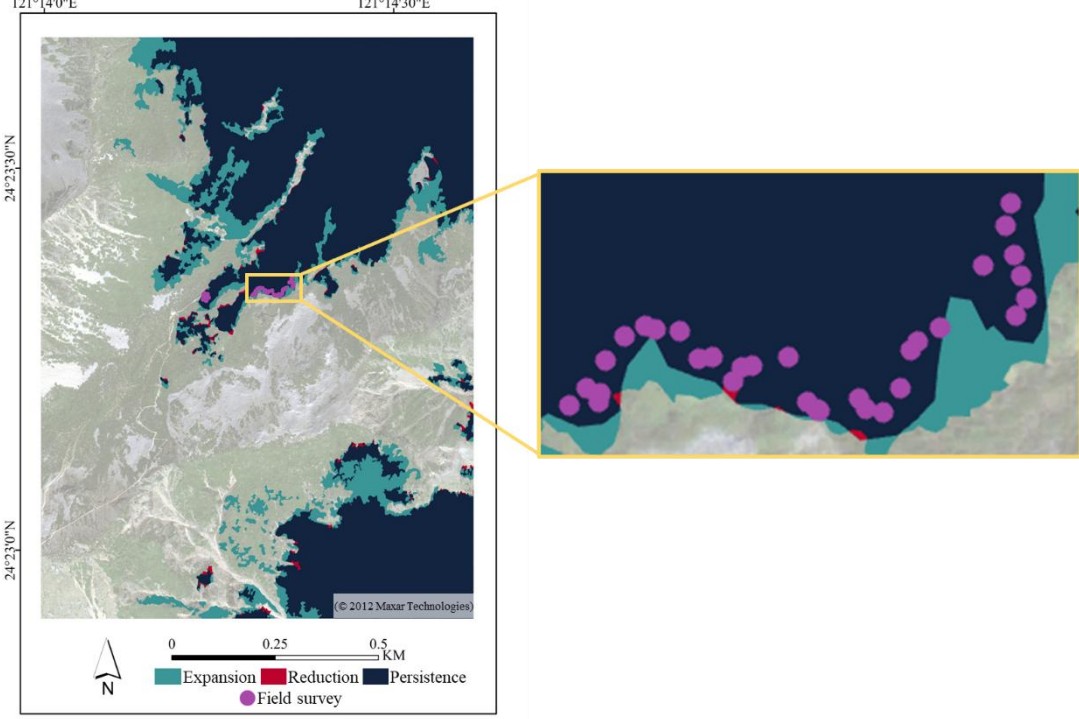

**Figure 8. The spatial distribution of ATE area changes from 2012 to 2021. ATE expansion is marked in dark cyan, reduction is marked in dark red, persistence is marked in dark blue, and field survey point in purple.**

**Table 8. Forest area, expansion, and reduction across different elevation from 2012 to 2021. The table includes forest area in 2012, net changes in area, and corresponding percentage changes.**

| Elevations (m) | Forest Area in 2012 (ha) | Expansion area (ha) | Reduction area (ha) | Net Change (ha) | Change (%) |
|---|---|---|---|---|---|
| 3300~3400 | 6.99 | 0.28 | 0.03 | 0.25 | 3.6 |
| 3400~3500 | 12.43 | 2.21 | 0.08 | 2.13 | 17.1 |
| 3500~3600 | 8.40 | 5.10 | 0.23 | 4.87 | 58.0 |
| 3600~3700 | 3.26 | 2.88 | 0.06 | 2.82 | 86.4 |
| 3700~3800 | 0.78 | 0.02 | 0.00 | 0.02 | 2.5 |

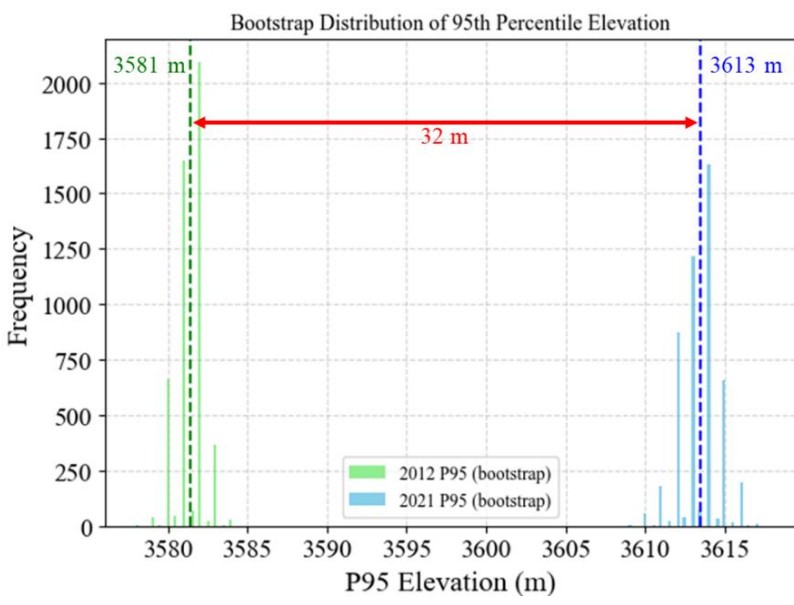

**Figure 9. Bootstrap distribution of the 95th percentile elevation of forest cover for 2012 and 2021. The histogram shows the frequency of estimated 95th percentile elevations (P95) based on resampling. Green bars represent 2012 estimates, while blue bars represent 2021. The dashed vertical lines indicate the mean P95 value for each year.**

## 4. Discussion

### 4.1 Comparison with previous alpine treeline ecotone remote sensing studies

Recent advancements in remote sensing technology have enabled extensive studies on alpine treelines using imagery at various spatial resolutions (Garbarino et al., 2023). For example, Xu et al. (2020) employed Landsat (30 m) data to assess treeline–climate relationships in China, reporting an upward shift of ~50 m per 1°C increase in temperature. At medium to high resolution, Rösch et al. (2022) achieved over 90% classification accuracy for *Pinus mugo* in the Alps using PlanetScope (3 m) and Sentinel-2 (10 m) data, emphasizing the value of multi-source data fusion. At very high resolution, Terskaia et al. (2020) combined aerial orthophotos (1–2 m) and WorldView-2 imagery (0.5 m) to quantify shrub and tree encroachment in Alaska, detecting substantial vegetation transitions over six decades.

Building on prior work, fine-scale mapping of alpine treeline ecotones (ATEs) remains difficult because transitional vegetation is spatially heterogeneous, often includes stunted or shrubby forms such as krummholz, and exhibits subtle spectral/structural gradients at meter scales (e.g., Bader et al., 2021; Nguyen et al., 2022). Our study uses ultra-high-resolution WorldView-2 imagery (0.4 m) and machine learning workflows to detect fine-scale transitions within the ATE (~400 ha) in Taiwan. Concretely, we show that integrating spectral bands, vegetation indices, and texture (GLCM) features at sub-meter resolution enables reliable separation of krummholz from closed-canopy forest—an underrepresented class distinction in many alpine studies (cf. Korznikov et al., 2021; Nguyen et al., 2022). This demonstrates the novelty and practical value of combining modern machine-learning segmentation with ultra-high-resolution imagery to fine-scale analyze the alpine treeline ecotone (ATE) in subtropical mountain environments. Related recent work similarly highlights the need for meter-scale approaches to capture ATE patterns and dynamics (Zou et al., 2022; Carrieri et al., 2024).

### 4.2 Alpine treeline ecotone changes and spatial patterns

Our findings reveal that, from 2012 to 2021, the forest class of alpine treeline ecotone (ATE) in the Xue Mountain glacial cirque shifted upward by $32.00 \pm 4.00$ meters, accompanied by a pronounced densification of forest cover. This finding aligns with patterns observed in other mountainous regions worldwide. For example, in Taiwan's Hehuan Mountain and Yushan, similar upward shifts in treeline position and increases in forest density have been reported (Greenwood et al., 2014; Chung et al., 2021). Likewise, Davis et al. (2020) observed an upslope advance of $0.83 \pm 0.67$ m/year for several tree species in the Rocky Mountains of Canada. In contrast, studies in the European Alps have noted significant reductions in snow cover and increased alpine vegetation productivity, potentially enhancing local carbon sequestration, although with a limited global impact (Rumpf et al., 2022). Additionally, in the eastern Himalayas, over 80% of trees have already reached the thermal treeline, with projected upslope migration of 140 meters by the end of the 21st century due to warming (Wang et al., 2022). These comparisons support the robustness of our observed treeline ecotone dynamics and highlight both global consistency and regional variation in alpine ecosystems response to climate change. It should be noted that the temporal comparison was limited to the ~150 ha cloud-free overlap between 2012 and 2021 imagery, which may slightly underestimate the total forest expansion within the broader 400 ha study area.

Despite the overall satisfactory classification performance, some confusion between forest and krummholz was observed due to their similar canopy structures and spectral reflectance. This misclassification occurred mainly along the transition between dense forest to stunted krummholz. However, this issue had only a limited influence on the overall outcomes. Field survey validation confirmed that the classified treeline boundaries were consistent with the observed forest–krummholz transitions in situ, and both RF and U-Net models maintained high accuracies (OA > 0.83, Kappa > 0.76). Therefore, the local confusion slightly affected boundary precision but did not alter the overall trend of the alpine treeline ecotone. To further minimize this effect in future work, incorporating structural features, such as LiDAR-derived canopy height models, could improve discrimination between forest and krummholz and enhance classification reliability.

Regarding alpine treeline ecotone spatial patterns, Bader et al. (2021) classified alpine treeline patterns into discrete and gradual categories, further distinguishing them into gradual, diffuse, abrupt-diffuse, abrupt, and tree island treelines. Based on the classification results derived from high-resolution satellite imagery, this study identified the treeline patterns in the Xue Mountain glacial cirque as abrupt and tree island treeline patterns. However, additional long-term field observations are required to further investigate the underlying treeline dynamics and demographic processes (Liao et al., unpublished data).

### 4.3 Feature importance

In this study, a total of 77 features were derived from the satellite imagery, including 8 spectral bands, 13 vegetation indices, and 56 texture features. To improve model efficiency, we ranked features using the Random Forest (RF) model and selected the top 61 features, which accounted for 95% of the cumulative importance. Among them, SEVI, Yellow (Y), Blue (B), Green (G), and NDVI2 were identified as the most important for classifying the treeline ecotone. Notably, most of the top-ranked features were spectral or vegetation index variables, whereas texture features contributed less to the classification. The feature selection slightly improved the overall accuracy (+0.4%) and the Kappa coefficient. Although OA was used as the primary selection criterion, the analysis also confirmed that the selected features maintained or improved F1-scores for the forest class, the primary focus of detecting treeline changes. It should be noted that optimizing overall accuracy (OA) values may sometimes overlook minority or ecologically important classes. Therefore, we specifically examined the F1-score for the forest class—our primary concern for treeline detection—and verified that its classification performance was not compromised. This indicates that our feature selection strategy effectively balanced overall model performance with the accuracy of the most ecologically relevant land-cover category.

Although the numerical improvement in overall accuracy appears modest, such enhancement is ecologically meaningful. Even slight gains in classification precision can improve the detection of subtle land cover transitions, particularly the identification of forest expansion boundaries in alpine treeline ecotones. These improvements strengthen the ecological interpretation of spatial change dynamics and provide a more reliable foundation for long-term monitoring (e.g., Bader et al., 2021; Wang et al., 2022).

These findings align with previous studies on vegetation classification using multispectral satellite imagery, though the most informative spectral bands may vary depending on the sensor, study region, and forest type. For instance, studies using Sentinel-2 imagery (10–20 m resolution) found the shortwave infrared (SWIR), red, and near-infrared (NIR) bands to be particularly effective in forest classification tasks. Bolyn et al. (2018) identified SWIR, red, and NIR as the most important features for classifying forest types, while Immitzer et al. (2019) emphasized the role of red and NIR in time-series-based tree species mapping. Similarly, Hościło and Lewandowska (2019) reported improved forest type discrimination when using multi-temporal red, NIR, and red-edge bands. In contrast, studies using WorldView-2 imagery (high-resolution, 0.4–1.6 m) revealed different key spectral bands. Abutaleb et al. (2021) found that the green, yellow, red, and NIR2 bands were most relevant for mapping eucalyptus trees in a subtropical environment. On the other hand, Immitzer et al. (2012) reported that blue, green, red, and NIR1 bands were particularly effective in classifying coniferous forest types in Austria.

These variations underscore the contextual nature of feature importance, suggesting that optimal band selection depends on factors such as spatial resolution, vegetation structure, and topographic complexity. Our results—emphasizing SEVI, Y, B, G, and NDVI2 —are well-suited to the alpine treeline ecotone in Taiwan, where coniferous species such as *Abies kawakamii* dominate.

### 5. Conclusions

This study investigates changes in the alpine treeline ecotone (ATE) of the Xue Mountain glacial cirques in Taiwan from 2012 to 2021, utilizing WorldView-2 imagery integrated with Random Forest and U-Net models. The alpine treeline ecotone

(ATE) in Xue Mountain glacial cirques was a transitional ecotone where krummholz species—such as Yushan juniper (*Juniperus morrisonicola*) and Yushan rhododendron (*Rhododendron pseudochrysanthum*)—begin to appear within the alpine talus slope. By incorporating spectral bands, vegetation indices, and texture features, we achieved high classification accuracy and computational efficiency for detailed delineation, supported by both satellite classification results and GNSS-referenced field survey data. The classification results could provide a basis for further analysis, including ATE patterns, microenvironment conditions, and how vegetation interacts with the microenvironment under climate change scenarios. Feature selection identified the most important variables as the Shadow-Eliminated Vegetation Index (SEVI), Yellow (Y), Blue (B), Green (G) bands, and Normalized Difference Vegetation Index (NDVI2), which can serve as key information for forest management and monitoring. Over the past decade, the study area gained approximately 10.09 hectares of forest cover, indicating that trees grew, canopies became denser, or saplings increased. Additionally, the upper limit of forest distribution shifted upslope by $32.00 \pm 4.00$ meters, revealing that forests expanded to higher elevations. These findings offer new insights into ATE dynamics in Taiwan's alpine environment and demonstrate the potential of integrating machine learning techniques with high-resolution satellite imagery for long-term ecological monitoring.

**Data availability**

Data are available upon request from the corresponding author (Hui Ping Tsai).

**Author contributions**

Conceptualization, GGW, MCL, WW, HPT and HYT; methodology, GGW, MCL, WW, HPT and HYT; software, GGW, and HPT; validation, GGW, MCL, WW, HPT and HYT; formal analysis, GGW, HPT and HYT; investigation, GGW, MCL, WW, HPT and HYT; resources, MCL, HPT and HYT; data curation, GGW, MCL and WW; writing—original draft preparation, GGW and HPT; writing—review and editing, MCL, HPT and HYT; visualization, GGW and HPT; supervision, HPT and HYT; project administration, HPT and HYT; funding acquisition, MCL, HPT and HYT. All authors have read and agreed to the published version of the manuscript.

**Competing interests**

The authors declare that they have no conflict of interest.

**Acknowledgments**

This work was partially financially supported by the "Innovation and Development Center of Sustainable Agriculture" from The Featured Areas Research Center Program within the framework of the Higher Education Sprout Project by the Ministry of Education (MOE) in Taiwan. Additionally, the support also provided by the National Science and Technology Council under projects 111-2313-B-054-001, 112-2321-B-005-007-, 112-2634-F-005-002-, 112-2119-M-005-001-, 112-2121-M-005-003-, 113-2321-B-005-005-, 113-2634-F-005-002-, 113-2119-M-005-001-, 113-2121-M-005-005- and Shei-Pa National Park Headquarters, National Park Service, Ministry of the Interior under project SP113110.

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
