# Peer review of "Machine learning-based Alpine treeline ecotone detection on Xue Mountain in Taiwan"

_EGUsphere, 2025_

## Author Comment (AC1)

Responses to Reviewer #2

Dear authors,

It was a pleasure to read your well-written manuscript. You used two machine-learning (ML) methods to classify four land-cover/ image section classes (forest, krummholz, bare ground and shadow) in a montane-alpine transition in Taiwan, repeating the classification for images from two years to detect changes through time.

The machine-learning methods are very nicely explained. However, I am missing information about how you obtained your "ground truth". This is actually not collected on the ground or by manual/visual labelling, if I interpreted the flowchart correctly, but by automated (?) pixel-based classification (not further specified…). So it sounds like you use one classification method to validate another, which seems strange and a bit circular. Why did you not just use the pixel-based classification for both images, if that worked so well that you could use it as "ground truth"? Why the step of developing the ML methods?

**Response:** We thank the reviewer for his insightful question. The ground truth labels in our study were not derived from automatic classification or machine learning output but were confirmed through image digitization and expert ground surveys. These manually annotated labels can be used as independent ground truth for training, validating, and testing machine learning models. We revised the manuscript as follows:

"Ground truth data for the study area were manually labeled using a pixel-based approach and categorized into four classes"

Pixel-based labels are used solely for model training and evaluation and not for generating classification results. Due to the complexity of the alpine treeline ecotone, spectral signals are often affected by shadows, terrain, and mixed vegetation, making traditional pixel-based or threshold-based methods inaccurate. Therefore, we adopted random forest and U-Net models, which can integrate spectral and spatial features (including vegetation index and texture indicators), enabling more robust and general classification in two periods (2012 and 2021). This approach has higher sensitivity to spatial structure and reduces classification errors in complex areas such as transition zones or shadow areas.

It appears to me that very typical patterns of "shade" and "forest" are produced, that should allow a ML model to recognize forest at a somewhat larger scale than at the pixel level. Of course, if you train your model with a pixel-level classification of spectral signatures, the ML model is going to reproduce this, but if you would use real ground-truth data or manually labelled forest areas to train the model, it may be able to really recognize forest and to make use of the shadow rather than to have it only as a nuisance (it will still be a nuisance where whole hillslopes are in shadow, but within the forest, it could become part of the signal, and should be capturable in the texture variables, perhaps if you use a different scale (number of neighbours) to calculate the texture.

**Response:** Thanks for your comment. We noted that shadows are often associated with canopy gaps in continuous forest areas. In our study area, shadows also occur near adjacent scree slopes, representing terrain characteristics.

Based on the observation of orthorectified aerial imagery and nearby field survey, we would like to classify shadows as a separate class.

Another important piece of information that you need to elaborate upon is how you defined the treeline/ treeline ecotone, in the field and on the classified images, what the survey data consist of, and how you compared the survey data to the treeline location suggested by the classification.

**Response:** Thanks for your comment. We define the treeline ecotone not as a fixed linear boundary but as a transitional zone where krummholz, such as Yushan Juniper (*Juniperus morrisonicola*) and Yushan rhododendron (*Rhododendron pseudochrystam*), begin to appear within the alpine talus slope (Liao. 2016; Liao et al., 2023).

Reference:

Liao, M. C. Vegetation Structure of Subalpine Ecosystem in Taiwan: A Case Study of Xue Mountain. PhD Thesis. Taichung City. Taiwan. 2016

Liao, M. C. Wang, W., and Tzeng H. Y.: Study of the Structure and Competitive Coexistence of Subalpine Krummholz Species in Taiwan. Taiwan J. For. Sci., 38(3), 203-220, https://doi.org/10.7075/TJFS.202309_38(3).0002, 2023.

Usually, the accuracy of ML models is much higher within the image it was trained on that on new images, since these may differ in e.g. lighting, season, angle, etc. Figure 7 a and c show that indeed the lighting seems to have been quite different in the two images. Therefore, it is not unlikely, that the classification accuracy was a lot lower for the year not used for model training, so that part of the differences through time could be due to this different accuracy. Did you validate the accuracy for the other year in any way? It would be important to do and show a detailed manual (not based on another automated classification method) validation of the results, seeing whether and where the forest, in particular, is classified correctly, especially at the boundary between forest and non-forest, i.e. in the treeline ecotone. You could e.g. use imagery from Google Maps or Bing, which have a higher spatial resolution for many parts of the world than Worldview images and are often available for past dates, or perhaps there are aerial images available from more local sources. These would allow you to check whether the areas where you detected change really appear to have changed in reality. I guess your field survey data might also show this, but it is currently unclear from the manuscript how these were used. Related to this, especially if you define forest elevational shift by the single highest point, there is a reasonable chance that a change in that point does not represent a real shift. Perhaps you can think of a different, more robust, measure of treeline-ecotone elevation?

**Response:** Thank you for your comments. We examined the imagery using Google Earth Pro; however, we found that the inter-annual differences in our study area were often subtle and unclear. Therefore, we relied on high-resolution satellite imagery (WorldView) to investigate ATE change in detail.

Our ground truth data were not derived from automated classification outputs. Still, they were independently generated through manual digitization and expert visual interpretation of the satellite imagery, further cross-validated through proximity to field survey data. These ground truth polygons were used for training and validating the classification, ensuring independence from the model and robustness in assessing accuracy.

As mentioned, we attempted to utilize Google Earth Pro and other high-resolution base layers for visual validation. However, due to limitations in temporal availability and clarity for our mountainous region, these sources could not be reliably to identify changes. We will clarify this limitation in the revised manuscript and emphasize our reliance on consistent and high-resolution satellite data.

We appreciate your suggestion to adopt a more robust measure for treeline elevation rather than relying solely on the single highest forest pixel. In response, we applied a bootstrap resampling method (5,000 iterations) to estimate the distribution of treeline elevations. This approach allowed us to calculate a statistically robust mean elevation change. Our analysis shows that the treeline has shifted upward by 32.00 meters from 2012 to 2021, with a 95% confidence interval of ±4.00 meters. We revised the manuscript by adding one figure (Figure 9).

[Figure]

Figure 9. Bootstrap distribution of the 95th percentile elevation of forest cover for 2012 and 2021. The histogram shows the frequency of estimated 95th percentile elevations (P95) based on resampling. Green bars represent 2012 estimates, while blue bars represent 2021. The dashed vertical lines indicate the mean P95 value for each year.

About the presentation: the figures are well-prepared. I like figure 2, in particular, as it nicely explains the different steps taken in the research and the connection between them. However, the captions for all of the figures and tables are much too short. They do not explain what is in the respective figure or table. Please expand and try to make the figures and tables self-explanatory, i.e. if a reader goes to look at them without having read the main text, they can sort of understand what they are about. See my example below for Fig 1. I also would advise not to use title font (capitalized words) in the figure captions or section titles.

**Response:** Thank you for your comments. We revised all captions for all figures and tables in the new version of the manuscript.

**Some more detailed comments:**

You sometimes report more decimal numbers than is reasonable or useful (e.g. L 21, 24, 48, 49) Please check for this and reduce the unnecessary precision

**Response:** Thank you for the comment. We revised the numerical values to display no more than three decimal places in the manuscript.

Avoid sentences like "The F1-score results are shown in Fig. 4." – instead, give the results and then just cite the figure: The result was X (Fig, 4).

**Response:** Thank you for the comment. We revised the manuscript.

**Title**: Change Alpine to alpine (the capital letter would suggest that you worked in the Alps, whereas the small a is used for alpine as a life zone in general)

**Title**: on Xue Mountain in Taiwan / in the Xue Mountains of Taiwan

**Response:** Thank you for the comment. We revised the title of manuscript as follows:

"Machine learning-based alpine treeline ecotone detection on Xue Mountain in Taiwan"

**Abstract:**

L14 do not use capitals for alpine treeline ecotone

L18 remove "the" before Random Forest (and not sure that random forest needs to be capitalized)

L19 & 26-27 either use the introduced abbreviation (ATE) or the full term (alpine treeline ecotone), but not both every time

**Response (L14, L18, L19 & 26-27):** Thank you for the comment. We revised these sentences.

**Introduction**

L33-34 "Alpine zone ecosystems are susceptible to environmental changes compared to other regions": are they really? Maybe they are not, because of the high environmental heterogeneity and small migration distances, compared to latitudinal climate gradients…

**Response:** Thank you for the comment. We revised the manuscript as follows:
"Alpine zone ecosystems are particularly vulnerable to environmental change due to their high environmental heterogeneity and limited species migration distances, especially when compared to broader latitudinal climate gradients and more resilient lowland regions."

L35-36 I would suggest referring to this transition as the alpine treeline ecotone (not alpine treeline and not capitalized). This is partly a matter of habit and taste, but it was recently suggested to reserve "treeline" for the climatic potential, and treeline ecotone to the actual observed transition from forest to alpine (or upper forest limit if it is unclear whether the transition is even related to climatic limitations) See e.g. Körner & Hoch 2023 and Malanson 2024. If you decide to follow this terminology, check its use throughout the manuscript.

**Response:** Thank you for the comment. We consociated our study terminology with the alpine treeline ecotone (ATE) and revised the manuscript accordingly.

L38: Bader et al., 2020 should be 2021

**Response:** Thank you for the comment. We corrected it.

L38-39 "Furthermore, ATE changes illustrate the impact of climate change…" Why "furtherore"? And how do you know that climate change is the driver of change? Are there no other potential drivers?

**Response:** Thank you for the comment. You are right that climate change is indeed a major driving force behind changes in alpine treeline ecotone (ATE), but research shows that there are other potential factors that also influence changes in treeline location and structure. We revised this sentence as follows:

> "Changes in the alpine treeline ecotone (ATE) illustrate the impacts of climate change on mountain ecosystems, such as the upward migration of tree species and increased tree density. However, these shifts are also influenced by other drivers, including land-use history, altered disturbance regimes (e.g., fire disturbance), herbivory pressure, and species-specific physiological traits (Wang et al., 2016; Johnson et al., 2017; Du et al., 2018; Mohapatra et al., 2019)."

L42 "to study alpine treelines." You could cite Garbarino et al 2023 here

L41-54 This paragrpahs gives some examples, but the seem a bit of a random pick. Can you highlight how they are somehow connected (e.g. three examples of studies at different spatial resolutions (please provide the sensor resolution for each data source used), three examples of change detection, or something else.

L52-53 These percentage have no meaning, since we do not know what the reverence area was.

L54 Careful, these studies do not tell us anything about the reliability of the methods applied. Maybe the confirm the usefulness or the great potential or something like that, but not the reliability

**Response (L42, L41-54, L52-53, L54):** Thank you for the comment. We reorganized the content by grouping the cited studies according to spatial resolution and sensor type, clarified the reference areas for percentage changes where applicable, and revised the concluding sentence to avoid overstating the reliability of remote sensing methods. These changes aimed to improve both scientific accuracy and narrative coherence.

L56 "favorable results" of what? Classification accuracy?

**Response:** Thank you for the comment. We revised "vaforable results" to "promising classification results".

L58 municipality

**Response:** Thank you for the comment. We corrected it.

L62 the tree Cecropia hololeuca, which has a optically striking shape and colour (I think, check the original paper to confirm)

**Response:** The tree species *Cecropia hololeuca* exhibits distinctive spectral signatures associated with its large leaves and crown structure. Furthermore, due to the abundance of *Cecropia hololeuca*, their visually prominent shape and bright gray coloration make them readily identifiable to the human eye in RGB imagery.

Reference:
Wagner, F. H., Sanchez, A., Tarabalka, Y., Lotte, R. G., Ferreira, M. P., Aidar, M. P. M., Gloor, E., Phillips, O. L., and Aragão, L. E. O. C.: Using the U-net convolutional network to map forest types and disturbance in the

Atlantic rainforest with very high resolution images. Remote Sens. Ecol. Conserv., 5(4), 360-375, https://doi.org/10.1002/rse2.111, 2019.

L63 "The classification accuracy for Cecropia hololeuca species reached 97%, with an IoU of 0.86." This is a bit too much technical detail at this point.

L68-69 "Based on these studies, applying…" à Based on these studies, we conclude that applying…

**Response (L63, L68-69):** Thank you for the comment. We revised these sentences.

L80 with "studies on the volume estimation", do you mean "studies estimating wood volumes"?

L81 It is not clear here whether these studies were done in the alpine treeline ecotone, or in the forest below.

**Response (L80, L81):** Thank you for the comment. We revised these sentences as follows:

"Most ecological studies conducted in this research area have focused on Taiwan fir forests, with several researchers estimating wood volumes, competitive pressure, forest structure, and spatial distribution of the species primarily through field surveys conducted below the alpine treeline ecotone."

L82 remove the sentence referring to the figure (just add (Fig 1). Instead please explain here what "krummholz" means to you. How do you define it? And the same for "forest" This is quite relevant in a treeline context.

**Response:** Thank you for the comment. We defined krummholz and revised the manuscript as follows:

"Ground truth data for the study area were manually labeled using a pixel-based approach and categorized into four classes: (1) Bare land, referring to areas of exposed soil, rock surfaces, or sparsely vegetated ground; (2) Forest, defined as regions with dense, continuous tree canopy cover; (3) Krummholz, representing stunted, shrub-like trees typically found at high elevations near the treeline and shaped by wind or snow pressure (Liao et al., 2023); and (4) Shadow, representing regions with low reflectance caused by topographic shading or solar angle effects."

Reference:
Liao, M. C. Wang, W., and Tzeng H. Y.: Study of the Structure and Competitive Coexistence of Subalpine Krummholz Species in Taiwan. Taiwan J. For. Sci., 38(3), 203-220, https://doi.org/10.7075/TJFS.202309_38(3).0002, 2023.

Figure 1 Please provide a self-explanatory caption. E.g. "Location of the treeline ecotone study area in the Xue Mountain glacial cirques in Shei-Pa National Park (top-right map) in north-central Taiwan (top-left map). The red marker in the aerial image (bottom-left map) indicates….. The digital elevation model shown in the bottom-right image shows the same area as the aerial image and covers the entire study area"  Instead of having to refer to the "top-left map" you could also add letters to each panel.

Thank you for the comment. We modified Figure 1 as follows:

[Figure]

Figure 1. Geographic location of the treeline ecotone study area in the Xue Mountain glacial cirques in Shei-Pa National Park (top-right map) in north-central Taiwan (top-left map). The red marker in the Worldview-2 image (bottom-left map) indicates research area. The digital elevation model shown in the bottom-right image shows the same area as the Worldview-2 image and covers the entire study area"

**Methods** & **results**: please write the methods and results sections in the past tense

**Response:** Thank you for the comment. We revised methods and results sections in the past tense.

L89 How was the optimal model selected, did you validate the classification at all? How did you obtain the "ground truth"- from the flow chart fig 2, it looks like you obtained it from the image itself, rather than from survey data? Survey data are not in the flow chart at all.

L93 I suggest writing "is reported to be within 3 meters"

**Response(L89, L93):** Thank you for the comment. We revised the manuscript.

Figure 2: This is a nice figure (but the caption needs to explain what this is a research flow for (e.g. "Research flow for classifying Worldview images of a treeline ecotone on Mt Xue in Taiwan for detecting changes in forest cover.").   A detailed question: in section 7, the two images that are subtracted indeed look a bit different, but in Figure 6, they look much more similar. What explains this discrepancy? Maybe in 6, b and c are accidentally the same image..?

**Response:** Thank you for the helpful comment. We revised the caption of Figure 2 as follows:

> "Figure 2. Research flow for classifying WorldView-2 images of a treeline ecotone on Mt. Xue in Taiwan to detect treeline changes. The process begins with WorldView-2 satellite image acquisition, followed by feature extraction (spectral bands, vegetation indices, and texture features), model training using Random Forest (RF) and U-Net, accuracy evaluation, feature selection, and temporal analysis of alpine treeline changes between 2012 and 2021.."

Regarding Figure 6, we appreciate your observation. After careful re-examination, we confirm that Figure 6(b) and Figure 6(c) are not the same image. The differences are subtle but can be observed, for example, changes in the forest area in the northwest and the shadowed region in the northeast. Specifically, Figure 6(b) shows the classification results using all 77 features, while Figure 6(c) presents the results after applying feature selection, which reduces the number of features to 61. The overall appearance is similar because the selected features retain most of the classification power, but minor differences remain in certain regions.

L104 "and GPS was used to record survey points" This addition seems out of place…

**Response:** Thank you for the comment. We revised this sentence as follows:

> "GPS devices were used to record field survey points, which were subsequently used to verify ATE positions and assist in manual ground truth labeling"

L107 2.4 Vegetation indices

**Response:** Thank you for the comment. We revised this word.

L115-116 This reads like it came out of the research proposal. Here, please use the past tense.

**Response:** Thank you for the comment. We revised these sentences.

L149 "calculated using the following formula:" and then no formula follows…

**Response:** Thank you for the comment. We have revised this sentence.

L173, Fig 3 With "ground truth data" you mean manually classified images for training and validation? That is not really ground truth, is it? Maybe call this "labelled data" instead?

**Response:** Thank you for the comment. In this study, we continue to refer to the manually labeled data as "ground truth" because these labels were created through a careful pixel-based manual classification process and serve as the reference for both training and validation. While we understand that "ground truth" can be interpreted differently, in remote sensing and image classification literature, it is common to use the term to describe manually labeled data used for validation and accuracy assessment.

L175 Can you define these classes? For example what is "bare land"? Rocks? Soil? Does it include alpine vegetation other than krummholz?

**Response:** Thank you for the comment. We defined and revised the manuscript as follows.

> "Ground truth data for the study area were manually labeled using a pixel-based approach and categorized into four classes: (1) Bare land, referring to areas of exposed soil, rock surfaces, or sparsely vegetated ground; (2) Forest, defined as regions with dense, continuous tree canopy cover; (3) Krummholz, representing stunted, shrub-like trees typically found at high elevations near the treeline and shaped by wind or snow pressure (Liao et al., 2023); and (4) Shadow, representing regions with low reflectance caused by topographic shading or solar angle effects."

L176 "The dataset was randomly split, with 80% used for training and validation and 75% and 25% allocated for training and validation,"… something appears to be wrong here.

**Response:** Thank you for the comment. The dataset was randomly split. We revised this sentence as follows:

> "Specifically, 80% of the patches were randomly selected for training and validation (with a 75/25 split), and the remaining 20% were used as an independent test set."

L194 Explain what the F1 score is, E.g. the accuracy of the models can be depicted by the F1 score, which exceeded… etc.

**Response:** Thank you for the comment. We revised this sentence.

L195 what do you mean with "they tend to influence each other more"? That they are confused more often?

**Response:** Thank you for the comment. We revised this sentence as follows:

> "Forest and krummholz were more frequently misclassified with one another due to their similar vegetation structures, while bare land and shadow were more easily distinguished, achieving F1-scores above 0.8."

L199-203 It seems that you may be over-interpreting the differences in F1 scores between the combinations. Most differences seem quite minimal. I would suggest writing that the combinations performed similarly well, pointing only at the larger differences that may actually mean something in terms of model performance (for the U-net model). It would also help to remind the reader what features are included in what combination, as this teaches us something about the importance of e.g. texture for recognizing different cover classes.

**Response:** Thank you for the comment. We revised this paragraph as follows:

> "Overall, the different feature combinations produced similar classification performance, with only minor differences observed across classes and models. In the RF model, bare land and shadow achieved the highest F1-scores (0.905 and 0.866, respectively) when using Combination 1 (spectral bands only). Forest and krummholz performed slightly better with Combination 4 (spectral bands, vegetation indices, and texture features), achieving F1-scores of 0.827 and 0.776, respectively. In the U-Net model, Combination 1 yielded the best result for bare land (F1 = 0.889), while Combination 4 slightly improved the classification of forest (0.828), krummholz (0.886), and shadow (0.869). These findings suggested that incorporating vegetation indices and texture features improved model performance for specific vegetation classes, particularly in the U-Net model, although overall improvements remained relatively modest."

L204 You just presented some accuracy metrics, and here you suddenly present other accuracy metrics; that is a bit confusing. Maybe ad "overall accuracy" (as opposed to the class-wise F1 scores), but I recommend just presenting the results here ("Similar to the accuracy patterns for the individual classes, in the U-Net models, the overall classification accuracy improved as the number of features increased, whereas for the RF model this was not the case (Table 5)"), not the table as such.

**Response:** Thank you for the comment. We revised this sentence as follows:

"The overall accuracy (OA) and Kappa coefficient for each feature combination (Table 5)."

Figure 4 Pleas repeat here what the different feature combinations were

**Response:** Thank you for the comment. We redrew Figure 4.

[Figure]

Figure 4. F1-scores for four land cover classes (forest, krummholz, bare land, shadow) using RF and U-Net models with different feature combinations."

Table 5 Explain abbreviations and combinations and number in brackets and bold font in the caption

**Response:** Thank you for the comment. We revised the Table 5 as follows:

"Table 5. Evaluation of classification accuracy using different feature combinations and models. Overall accuracy (OA) and Kappa coefficient are shown for Random Forest (RF) and U-Net models. Numbers in parentheses indicate the number of input features. Bold values indicate the best results for each metric."

L214 "The treeline is determined based on the boundary between bare land, forests, and krummholz." – I suggest using "The upper limit of the treeline ecotone", and explaining better how you define this line, since now there are three classes, and those will all three be heterogeneously spread across the landscape….Since treeline is defined by trees and the ecotone is generally also described by the patterns of tree cover, the accuracy of forest detection seems to be the most important, so in line L221 you could add behind "texture features were relatively less important" ", as also suggested by the low F1 scores for combination 3 (spectral + texture; Fig 4), although for forest, in particular, texture strongly increased the classification accuracy relative to providing just spectral information, but vegetation indices increased it more".

**Response:** Thank you for the comment. We revised this paragraph as follows:

"The upper limit of the treeline ecotone was determined based on the spatial distribution boundary where patches of forest transitioned into krummholz and bare land."

"In contrast, texture features were relatively less important overall, as also suggested by the low F1 scores for combination 3 (spectral, texture; Fig. 4). However, for the forest class in particular, texture

features significantly improved classification accuracy compared to using only spectral bands, although the inclusion of vegetation indices contributed even more to the performance.".

L218 remove the sentence that introduces Fig 5, just add (Fig 5)

**Response:** Thank you for the comment. We revised this sentence.

L221 remove the line break, it does not look like a new paragraphs should start here. If anything, start it with Using these 61…

**Response:** Thank you for the comment. We revised this paragraph.

L222 Why is it important to reduce training time? Is it more important to reduce training time than to get a better model fit? I guess this could become important if one would want to apply the model more broadly, but for your particular application it does not appear to matter. On the other hand, there is the concept of parsimony in model selection, i.e. select the model with the best fit, but penalizing for model complexity, i.e. make the model as complex as necessary, but no more. Is this concept related, does it apply to machine-learning models? If the last features add 5% of accuracy, that is the same order of magnitude as you effect size (the change in forest cover), so those 5% may be relevant…? You see, I am a bit confused. Perhaps it needs a bit more explanation why you decided to use feature selection.

**Response:** Thank you for the comment. Reducing the number of features not only shortened the computation time but also improved performance. After feature selection, the number of features was reduced from 77 to 61. The classification accuracy on forest classification was stable, but the classification accuracy for krummholz improved by 2%, which is essential for ATE interpretation. This follows the principle of parsimony, keeping the model simple while maintaining prediction capability.

L223 Remove the "additionally. That difference is not a difference…

**Response:** Thank you for the comment. We have revised this sentence.

Table 6 Could you provide the training time in hours, so the it is easier to understand the order of magnitude?

**Response:** Thank you for the comment. We have revised the table 6.

Figure 6 These maps are not so informative. Maybe one bigger map with the classes and including missclassifications would show better how good the classification is. The "Ground truth" looks like another automatic classification, please explain well in the methods how you got to this map, and maybe call it something other than "ground truth"…

**Response:** Thank you for the comments. The "ground truth" labels in our study were not derived from automated classification but from manual image digitization confirmed by expert field surveys. Additionally, we will enlarge and revise Figure 6 to better highlight the classification results and their differences, making the comparison more informative.

Figure 7 Please print as big as possible, it is very hard to see anything on these images. Could you draw the boxes shown in b and d in the images in a and c, and remove the big white boxes with 1 and 2 (they block the view)? Also, the symbols for the field investigations are hard to see. It would also be more informative if not only the location, but also the vegetation types of the field survey points would be shown. As it is , it is unclear what the field survey data are. Please explain these data in the methods section and again briefly in the figure caption and in line 234-235. In any case, to be able to evaluate the fit, the images would need to be much bigger. You could also plot the fits of the field (real ground truth!) and the image classification in a separate graph.

**Response:** Thank you for the comment. We revised the Figure 7 and paragraph as follows:

"A U-Net model was trained using 61 selected features derived based on feature importance. The trained model was applied to classify satellite images from 2012 and 2021. The classification results were validated against field survey data collected in 2021, which recorded vegetation types and the position of the tree line along an elevational gradient. As shown in Fig. 7, the tree line derived from the classification closely aligns with the tree line identified through field survey."

[Figure]

Figure 7 Comparison of satellite imagery and classification results from 2012 and 2021. Panels (a) and (c) show high-resolution satellite images for 2012 and 2021, respectively. Colored boxes in these images indicate the enlarged areas shown in (b) and (d). Panels (b) and (d) present the classification results of the corresponding enlarged regions using a U-Net model trained with 61 selected features. Triangles mark field survey locations. The classification-derived treeline ecotone shows strong agreement with the tree line identified through field surveys."

L232 Decadal changes in the treeline ecotone

**Response:** Thank you for the comment. We revised this subtitle.

L235 avoid using tree line, stick to treeline or, even better treeline ecotone

L235 Since you have not explained how you define treeline, this statement is impossible to follow.

**Response(L235):** Thank you for the comment. We revised the manuscript, and we defined the treeline ecotone as follows:

> "We define the treeline ecotone not as a fixed linear boundary but as a transitional zone where krummholz, such as Yushan Juniper and Yushan rhododendron, begin to appear within the alpine talus slope."

L236 "Over a decade, the proportion of forest and shadow areas increased by 3.4% and 8.5%,...." Really? The proportion of shadow area increased in the last decade?? Obviously this is just a matter of lighting when the image was taken, so I recommend rephrasing this result.

**Response:** Thank you for your comment. Our classification results show that forest and shadow areas increased by 3.4% and 8.5%, respectively. Considering that shadow areas can be affected by lighting conditions at the time of image acquisition, the observed increase may not fully reflect actual land cover changes. In future work, we will further investigate the impact of lighting on classification results and refine our processing methods accordingly.

L239 how did you define the elevation distribution of the forest? The uppermost forest pixels?

**Response:** Thank you for this thoughtful question. To determine the elevation distribution of forests, we extracted the elevation values of all pixels classified as forests from the DEM. Instead of simply using the uppermost forest pixels, we defined the upper elevation limit as the 95th percentile of forest pixel elevations. To robustly estimate treeline elevation change and its uncertainty, we also used a bootstrapped resampling approach to calculate 95% confidence intervals.

Figure 8 caption: between 2012 and 2021. It would also be helpful to see the persistent forest cover here.

**Response:** Thank you for the comment. We revised the Figure 8 as follows:

[Figure]

Figure 8. The spatial distribution of forest area changes from 2012 to 2021. Forest expansion is marked in blue, reduction is marked in orange, and persistent is marked in green."

L238, Table 9: it may be better to express the area changes in e.g. ha, instead of km2, to get nicer numbers.

**Response:** Thank you for the comment. Table 9 has been updated to express area changes in hectares (ha) instead of square kilometers (km²), resulting in more intuitive values for the reader.

L240-241 "with the most significant changes occurring in the 3,500 to 3,600 m range." If this is where most of the treeline ecotone was, it would be worth mentioning this here

**Response:** Thank you for the comment. We revised the sentence as follows:

> "With the most significant changes occurring in the 3,500 to 3,600 m range, which corresponds to the primary treeline ecotone change zone in the Xue Mountain region."

L241 and/or L308 "In comparison, the most stable area was observed in the 3,700 to 3,800 m range." – explain here that hardly any forest was found here at any time.

**Response:** Thank you for the comment. We revised the sentence as follows:

> "In comparison, the most stable area was observed in the 3,700 to 3,800 m range, where minimal forest presence was detected in both 2012 and 2021."

Table 7: please also provide the forest area   in each belt in 2012 and the % change

**Response:** Thank you for your comment. We have revised Table 9 as suggested. However, since we have deleted the original Table 8, the current Table 9 is now renumbered as Table 8. Table 8 as follows:

Table 8. Forest area, expansion, and reduction across different elevation from 2012 to 2021. The table includes forest area in 2012, net changes in area, and corresponding percentage changes."

| Elevations (m) | Forest Area in 2012 (ha) | Expansion area (ha) | Reduction area (ha) | Net Change (ha) | Change (%) |
|---|---|---|---|---|---|
| 3300~3400 | 6.99 | 0.28 | 0.03 | 0.25 | 3.6 |
| 3400~3500 | 12.43 | 2.21 | 0.08 | 2.13 | 17.1 |
| 3500~3600 | 8.40 | 5.10 | 0.23 | 4.87 | 58.0 |
| 3600~3700 | 3.26 | 2.88 | 0.06 | 2.82 | 86.4 |
| 3700~3800 | 0.78 | 0.02 | 0.00 | 0.02 | 2.5 |

**Discussion**

L256-267 & 275-288 As a general advice, better start with you results and then contrast or align it with other studies, rather than the other way around. Now these paragraphs read a bit like a second introduction, again listing studies without any obvious logical connection between them. If you start with your results, you can use connections like "In contrast…", or "Likewise" to make a clearer connection between other people's findings and your results.

**Response:** Thank you for the comment. We revised the paragraph as follows:

> "Our findings reveal that, from 2012 to 2021, the alpine treeline ecotone (ATE) in the Xue Mountain glacial cirque experienced an upward shift of $32.00 \pm 4.00$ meters, along with a pronounced densification of forest cover. This finding aligns with patterns observed in other mountainous regions worldwide. For example, in Taiwan's Hehuan Mountain and Yushan, similar upward shifts in treeline position and increases in forest density have been reported (Greenwood et al., 2014; Chung et al., 2021). Likewise, Davis et al. (2020) observed an upslope advance of $0.83 \pm 0.67$ m/year for several tree species in the Rocky Mountains of Canada. In contrast, studies in the European Alps have noted significant reductions in snow cover and increased alpine vegetation productivity, potentially enhancing local carbon sequestration, although with a limited global impact (Rumpf et al., 2022). Additionally, in the eastern Himalayas, over 80% of trees have already reached the thermal treeline, with projected upslope migration of 140 meters by the end of the 21st century due to warming (Wang et al., 2022). These comparisons support the robustness of our observed treeline dynamics and highlight both global consistency and regional variation in alpine responses to climate change."

> "While OA was used as the primary selection criterion, we also confirmed that these top-ranked features maintained or improved F1-scores for the forest class, which is the primary concern in detecting treeline changes. Although overall accuracy (OA) served as the primary metric for feature ranking, we recognize that optimizing for OA may sometimes overlook minority or ecologically important classes. To address this, we specifically examined the F1-score for the forest class—our main target for treeline detection— and verified that its classification performance was not compromised."

L271 Was there any relationship between the treeline pattern and the local change? E.g. did abrupt treeline stay more stable than ones with tree islands? A bit more ecological interpretation of the patterns found would be interesting (e.g. relationship with topography).

**Response:** Thank you for the comment. Yes, based on our field surveys, there are topographic correlations between treeline patterns and local changes, which we are currently working on in another manuscript.

L275-288 Can you discuss here what the difference is between feature selectin based on the overall accuracy and feature selection based on the accuracy of your target land-cover class (which was obviously not shadow, but forest?

**Response:** Thank you for the comment. We revised the paragraph as follows:

> "While OA was used as the primary selection criterion, we also confirmed that these top-ranked features maintained or improved F1-scores for the forest class, which is the primary concern in detecting treeline changes. Although overall accuracy (OA) served as the primary metric for feature ranking, we recognize that optimizing for OA may sometimes overlook minority or ecologically important classes. To address this, we specifically examined the F1-score for the forest class—our main target for treeline detection— and verified that its classification performance was not compromised."

L292 Maybe do not mention shadow here, since that is more a no-data area than a land-cover class…

L294-295 These numbers are very technical for a conclusion section…

L296 "SEVI, Y, B, G, and NDVI2." Here, since readers may read the conclusions without having read the whole paper, you might want to explain the abbreviations.

L297 Again, this cannot be understood without an explanation about what the survey data are and how treeline was defined.

L299 denser or expanded?

L299 at higher elevations

**Response (L292, L294-295, L296, L297, L299):** Thank you for the comment. We revised the paragraph as follows:

"This study investigates changes in the ATE of the Xue Mountain glacial cirques in Taiwan from 2012 to 2021, utilizing WorldView-2 imagery in conjunction with Random Forest and U-Net models. By incorporating spectral bands, vegetation indices, and texture features, we achieved improved classification accuracy and computational efficiency. Feature selection identified the most important variables as the Shadow-Eliminated Vegetation Index (SEVI), Yellow (Y), Blue (B), Green (G) bands, and Normalized Difference Vegetation Index (NDVI2). The treeline was defined not as a fixed linear boundary but as a transitional ecotone where krummholz species—such as Yushan juniper (*Juniperus morrisonicola*) and Yushan rhododendron (*Rhododendron pseudochrysanthum*)—begin to appear within the alpine talus slope. This delineation was based on both satellite classification results and GPS-referenced field survey data. Over the past decade, forest cover in the study area expanded by approximately 0.101 km², indicating both denser canopy growth and outward expansion. In addition, the upper limit of forest distribution rose by $32.00 \pm 4.00$ meters, indicating an upslope shift of the treeline at higher elevations. These findings provide new insights into treeline dynamics in Taiwan's alpine environment and demonstrate the potential of high-resolution satellite imagery for long-term ecological monitoring."

References: please ident them (the hanging parts of each reference, rather than the main author name) to make them easier to navigate

**Response:** Thank you for the comment. We revised the manuscript.

**References cited**

Garbarino, M., D. Morresi, N. Anselmetto, and P. J. Weisberg. 2023. Treeline remote sensing: from tracking treeline shifts to multi-dimensional monitoring of ecotonal change. Remote Sensing in Ecology and Conservation **9**:729-742. https://doi.org/10.1002/rse2.351

Körner, C., and G. Hoch. 2023. Not every high-latitude or high-elevation forest edge is a treeline. Journal of Biogeography **50**:838-845. https://doi.org/10.1111/jbi.14593

Malanson, G. P. 2024. Inclusions and exclusions in treeline definitions. Journal of Biogeography **51**:54-56. https://doi.org/10.1111/jbi.14729

---

## Author Comment (AC3)

Responses to Reviewer #1

**Main Comment**

**Reviewer:** The authors state that they used "two cloud-free WorldView-2 orthorectified images with a spatial resolution of 0.4 meters, acquired on November 3, 2012, and September 26, 2021." However, they later clarify that only the panchromatic (PAN) band is available at this resolution, while they appear to use the color bands instead. This is unclear—did they use pansharpening? Please clarify which bands were actually used, at what resolution, and whether pansharpening was applied.

**Response:** We appreciate this critical observation. Our study obtained the 8-band multispectral WorldView-2 imagery, which originally had a spatial resolution of 1.64 m, along with the 0.41 m panchromatic image. We applied a pansharpening process to the multispectral bands to enhance spatial detail, resulting in an effective spatial resolution of 0.4 m for all used spectral bands. This pansharpened dataset was used for all feature extraction, including vegetation indices and texture features. We revised the manuscript as follows:

> "To enhance spatial detail, all multispectral bands were pansharpened using the corresponding high-resolution panchromatic band, yielding a uniform spatial resolution of 0.4 meters across all datasets used for feature extraction. The pansharpened multispectral imagery was the basis for deriving vegetation indices and texture features."

**Reviewer:** The origin of the training data is not clearly explained. The authors write: "Ground truth data in the study area were labeled using a pixel-based approach and categorized into four classes: bare land, forest, krummholz, and shadow (Fig. 3)." Does this mean an operator manually classified these images? If both images were already classified, what is the purpose of the complex processing workflow? Were both images used for training? If only one image was used for training, why would we expect the same classification accuracy to transfer to the second image, especially given possible environmental and seasonal differences?

**Response:** Yes, the ground truth was manually labeled and validated by trained operators and domain experts using a pixel-based approach (Fig. 3) in the 2021 WorldView-2 orthorectified images. The 2021 images were used for model training and validation, while the 2012 images were used for model evaluation. The classification results of 2012 and 2021 images were compared for temporal change analysis.

The purpose of the complex processing workflow was hoping to establish a universal model that can provide reliable alpine treeline ecotone classification. Only 2021 images were used for model training. We did not expect to see the same classification accuracy. Instead, we evaluated the 2012 image classification results carefully and the results were used to study the temporal and spatial changes.

The environmental and seasonal differences were minimized by using the autumn images when vegetation in the area had entered dormancy with less phenological presentation such as flowering or leaf flushing.

Additionally, we applied histogram matching during preprocessing to reduce radiometric and color inconsistencies caused by differences in lighting and atmospheric conditions. We revised the manuscript as follows:

"Two orthorectified, cloud-free WorldView-2 images acquired on November 3, 2012, and September 26, 2021, were obtained from RiChi Technology Co., Ltd. (New Taipei City, Taiwan). Both images were captured in the autumn season when vegetation had entered dormancy, minimizing the influence of phenological variability such as flowering. Histogram matching was applied to ensure radiometric consistency across the two images. In addition, GPS devices were used to record field survey points, which were subsequently used to verify treeline positions and assist in manual ground truth labeling."

**Reviewer:** Regarding training, the authors mention using 512x512 patches and then splitting the dataset. Is the train/test split done at the patch level or at the pixel level (within patches)? This distinction is important, as pixel-level splits can introduce data leakage, especially in spatially autocorrelated datasets.

**Response:** Thank you for pointing this out. The dataset was split at the patch level, not the pixel level. We revised the manuscript as follows:

"Each image (5380 × 4671 pixels) was segmented into 110 non-overlapping patches of 512 × 512 pixels. The dataset split was performed at the patch level, not the pixel level, to avoid spatial autocorrelation and data leakage. Specifically, 80% of the patches were randomly selected for training and validation (75% for training, 25% for validation), and the remaining 20% of patches were used as the independent test set. The number of patches used for training, validation, and testing was 66, 22, and 22, respectively."

**Reviewer:** The use of Random Forest (RF) for variable importance analysis is questionable. This approach is valid only if variables are independent, which is clearly not the case here. Additionally, is it worth performing this complex selection to save 20% of variables? Reducing from 77 to 61 features may not justify the effort, especially if interpretability or performance gain is marginal. As such, the entire discussion about variable importance remains inconclusive.

**Response:** Thank you for your comment. We acknowledge that Random Forest (RF) variable importance measures can be biased when input features are correlated. In our dataset, some of the 77 features—such as vegetation indices and texture metrics—are derived from overlapping spectral bands and are therefore not entirely independent. Nevertheless, RF remains a widely used method for feature selection in high-dimensional remote sensing and ecological data, and its robustness has been demonstrated even in the presence of correlated variables (Cutler et al., 2007; Belgiu & Drăguţ, 2016). In our study, RF was employed primarily to rank features and facilitate a conservative feature selection process, ultimately retaining the top 61 out of 77 features. This reduction resulted in a 14.3% decrease in training time and a slight improvement in overall accuracy (OA increased from 0.838 to 0.842). Although the gains were modest, this optimization was valuable considering the computational demands of the U-Net model.

Reference:

Cutler, D. R., Edwards Jr, T. C., Beard, K. H., Cutler, A., Hess, K. T., Gibson, J., and Lawler, J. J.: Random forests for classification in ecology. Ecology, 88(11), 2783-2792, https://doi.org/10.1890/07-0539.1, 2007

Belgiu, M., and Drăguţ, L.: Random forest in remote sensing: A review of applications and future directions. ISPRS J. Photogramm. Remote Sens., 114, 24-31, https://doi.org/10.1016/j.isprsjprs.2016.01.011, 2016.

**Reviewer:** Finally, the reported 14-meter height increase lacks context. The sentence "Forest area and highest point height difference from 2012 to 2021" is vague. Does this mean the authors extracted the maximum elevation value among all forest pixels? What was done to ensure robustness against outliers or noise? Also, scientific results are typically reported with associated uncertainties, which are missing here—or, if included, were not clear to me.

**Response:** Thanks for your comment. The 14-meter elevation gain was calculated based on the difference between the highest elevations of the forest area in 2021 and 2012. To elaborate on the finding, we have performed another analysis by investigating the elevation percentage of forest cover in 2021 and 2012. Based on the results, we revised the manuscript as follows:

> "Based on the 95th percentile of DEM elevation values of all pixels classified as forest (Fig. 9), the treeline showed an upward shift of 32.00 meters between 2012 and 2021. The 95% confidence interval (± 4.00 meters) was estimated using a bootstrap resampling method (5,000 iterations). Differences in area changes across various elevation ranges are detailed in Table 8, with the most significant changes occurring in the 3,500- to 3,600-m range. The most stable area was observed in the range of 3,700 to 3,800 m."

[Figure]

Figure 9. Bootstrap distribution of the 95th percentile elevation of forest cover for 2012 and 2021. The histogram shows the frequency of estimated 95th percentile elevations (P95) based on resampling. Green bars represent 2012 estimates, while blue bars represent 2021. The dashed vertical lines indicate the mean P95 value for each year.

**Reviewer:** Lastly, if the only interest was in changes to forest cover, why not classify the change directly instead of classifying each image independently?

**Response:** Thank you for the comment. We manually labeled each image. However, only the 2021 image was used for model training. The 2012 image was classified using the trained model, and the labeling for 2012 was done afterward for accuracy validation, not training. Direct change classification would require labels from both years or a different type of model. Our goal was to evaluate whether a model trained on recent data could still perform well on earlier imagery. Since both images were taken in autumn and we applied histogram matching, seasonal and lighting differences were minimized.

**Minor Comments**

"Taiwan has the highest density of high mountains globally, with over 200 peaks exceeding 3,000 meters in elevation."

→ This sounds too subjective. The result depends on the threshold chosen. I recommend rewriting as:

"Taiwan is one of the regions with the highest density of high mountains, with over 200 peaks exceeding 3,000 meters in elevation."

**Response:** Thank you for the comment. We revised the manuscript as follows:

"Taiwan is one of the regions with the highest density of high mountains, with over 200 peaks exceeding 3,000 meters in elevation."

The introduction goes beyond the immediate scope of the study. However, I appreciate that the authors took the time to place their work in a broader context.

"At the same time, the productivity of alpine treeline vegetation increased, enhancing the ability to sequester atmospheric $CO_2$ and mitigating the effects of climate change (Rumpf et al., 2022)"

→ If this is true, however it's also be stated that the global effect is likely minor. The sentence could be more balanced.

**Response:** Thank you for the comment. We revised the manuscript as follows:

"At the same time, the productivity of alpine treeline vegetation increased, enhancing the ability to sequester atmospheric $CO_2$ and mitigating the effects of climate change (Rumpf et al., 2022)"

Reference:
Rumpf, S. B., Gravey, M., Brönnimann, O., Luoto, M., Cianfrani, C., Mariethoz, G., and Guisan, A.: From white to green: Snow cover loss and increased vegetation productivity in the European Alps. Science, 376(6597), 1119-1122, https://doi.org/10.1126/science.abn6697, 2022.

---

## Referee Report (RR1)

The authors used ultra-high resolution for vegetation classification and change detection in alpine treeline ecotones, which holds certain scientific value, particularly by focusing on the identification of the krummholz category, a relatively under-researched area. The methods are reasonable and have practical application value in treeline studies. Overall, in the previous round of major revisions, the authors effectively addressed the reviewers' comments with high quality. I recommend accepting the paper after minor revisions.

Specifically, I have the following comments and suggestions:

The authors should address the generalizability of their methodology. This study was conducted in a not very large area, and it is unclear whether the classification process and input feature combination can be applied to larger or other regions. At a larger scale or for remote sensing of alpine treelines in other areas, some concerns may arise, such as whether cloud-free, ultra-high-resolution data can be obtained for most remote mountainous areas. Additionally, the selection of time periods may vary depending on the dominant species in the ecotone. This study area dominated by *Juniperus* and *Abies* species, but for widely distributed treeline species like *Larix* and *Betula*, the rationale for using autumn data may be questioned. While the authors may not need to add new validation process in this paper, it is recommended to briefly address these points in the discussion.

What is the specific purpose of placing the research area indicated by the red marker in the lower-left corner of Figure 1? Why not zoom in on the central part of the map? Alternatively, the label "*Mt. Xue main peak*" could be reduced in size to minimize unnecessary obstruction.

In L225, the statement "*All classes achieved F1-scores above 0.6*" seems somewhat redundant, as a F1-score of 0.6 is not a particularly strong benchmark. Moreover, based on Figure 4, it is clear that most F1-scores are above 0.7, with only one around 0.6. It is recommended to remove this sentence or replace it with the overall average F1-score.

L251, write out the full name of *ATE* as "*alpine treeline ecotone*." "*ATE*" itself is not a widely used abbreviation.

L269-271, The ecological significance reflected in the results can be moved to the discussion section, with relevant citations added to confirm the value of this minor classification accuracy improvement for ecological applications.

The readability of Figure 5 is poor, and the key points are not clear. It is recommended to enlarge the bar chart and highlight the values and ranking of the factors indicating their relative importance. The curve for cumulative model interpretability is not a highlight and does not need to be emphasized. It would be sufficient to label the factors corresponding to the 95% threshold only.

L291, "*expanded by 0.105 km² and was reduced by 0.004 km²*": Using km² as

the unit makes the values appear insignificant. If the authors intend to convey a significant trend of forest expansion, it is recommended to use hectares instead. Moreover, the unit in Table 8 is also hectares, so it is suggested to standardize the area unit throughout the paper (including the corresponding expressions in the abstract and the other sections).

In Figures 7 & 8, the "*field survey*" icon color is not very prominent. Recommended to change the color or add a black border to make it stand out more. Additionally, in Figure 8, it would be better to zoom in, as the pink triangle is hard to find now.

L364, similarly, provide the full name of "*ATE*" here.

---

## Referee Report (RR2)

[referee-annotated manuscript omitted]

---

## Author Response (AR2)

Responses to Reviewer #3

The authors used ultra-high resolution for vegetation classification and change detection in alpine treeline ecotones, which holds certain scientific value, particularly by focusing on the identification of the krummholz category, a relatively under-researched area. The methods are reasonable and have practical application value in treeline studies. Overall, in the previous round of major revisions, the authors effectively addressed the reviewers' comments with high quality. I recommend accepting the paper after minor revisions.

Specifically, I have the following comments and suggestions:

The authors should address the generalizability of their methodology. This study was conducted in a not very large area, and it is unclear whether the classification process and input feature combination can be applied to larger or other regions. At a larger scale or for remote sensing of alpine treelines in other areas, some concerns may arise, such as whether cloud-free, ultra-high-resolution data can be obtained for most remote mountainous areas. Additionally, the selection of time periods may vary depending on the dominant species in the ecotone. This study area dominated by Juniperus and Abies species, but for widely distributed treeline species like Larix and Betula, the rationale for using autumn data may be questioned. While the authors may not need to add new validation process in this paper, it is recommended to briefly address these points in the discussion.

**Response:** Thank you for this insightful comment. Specifically, the proposed classification framework combining WorldView-2 ultra-high-resolution imagery and U-Net/RF models can be adapted to other alpine treeline ecotones with cloud-free satellite data available. We also acknowledged that accessibility of imagery and species-specific phenology (e.g., Larix, Betula) may influence the optimal observation period. However, it's worth noting that neither genus, Larix nor Betula, is found in Taiwan's alpine treeline ecotone. We believe this is what makes Taiwan's alpine treeline ecotone unique.

What is the specific purpose of placing the research area indicated by the red marker in the lower-left corner of Figure 1? Why not zoom in on the central part of the map? Alternatively, the label "Mt. Xue main peak" could be reduced in size to minimize unnecessary obstruction.

**Response:** Thank you for the comments. We have modified Figure 1.

[Figure]

**Figure 1. Study area. (a)** Geographic location of Shei-Pa National Park in north-central Taiwan. **(b)** Treeline ecotone study area located in the Xue Mountain glacial cirques within Shei-Pa National Park. **(c)** WorldView-2 image showing the research area with topographic contours.

In L225, the statement "All classes achieved F1-scores above 0.6" seems somewhat redundant, as a F1-score of 0.6 is not a particularly strong benchmark. Moreover, based on Figure 4, it is clear that most F1-scores are above 0.7, with only one around 0.6. It is recommended to remove this sentence or replace it with the overall average F1-score.

**Response:** Thank you for the valuable suggestion. The sentence regarding "F1-scores above 0.6" has been deleted, and the section 3.1 and 3.2 has been rewritten.

L251, write out the full name of ATE as "alpine treeline ecotone." "ATE" itself is not a widely used abbreviation.

**Response:** Thank you for the comment. We have revised the manuscript.

L269-271, The ecological significance reflected in the results can be moved to the discussion section, with relevant citations added to confirm the value of this minor classification accuracy improvement for ecological applications.

**Response:** Thank you for the comment. The related ecological interpretation has been moved to the Discussion section 4.3 and supported with additional references (L364-368).

> "Although the numerical improvement in overall accuracy appears modest, such enhancement is ecologically meaningful. Even slight gains in classification precision can improve the detection of subtle land cover transitions, particularly the identification of forest expansion boundaries in alpine treeline ecotones. These improvements strengthen the ecological interpretation of spatial change dynamics and provide a more reliable foundation for long-term monitoring (e.g., Bader et al., 2021; Wang et al., 2022)."

The readability of Figure 5 is poor, and the key points are not clear. It is recommended to enlarge the bar chart and highlight the values and ranking of the factors indicating their relative importance. The curve for cumulative model interpretability is not a highlight and does not need to be emphasized. It would be sufficient to label the factors corresponding to the 95% threshold only.

**Response:** Thank you for the comment. We have modified Figure 5, and the updated version is now presented as Figure 6.

[Figure]

Figure 6. Feature importance ranking derived from the Random Forest model. Features are ranked based on their contribution to classification accuracy, with the top-ranked features including SEVI, Y (yellow), B (blue), G (green), and NDVI2. Most of the top features are spectral bands and vegetation indices, while texture features rank lower.

L291, "expanded by 0.105 km² and was reduced by 0.004 km²": Using km² as the unit makes the values appear insignificant. If the authors intend to convey a significant trend of forest expansion, it is recommended to use hectares instead. Moreover, the unit in Table 8 is also hectares, so it is suggested to standardize the area unit throughout the paper (including the corresponding expressions in the abstract and the other sections).

**Response:** Thank you for the comment. We have revised the manuscript to standardize the area unit throughout the paper.

In Figures 7 & 8, the "field survey" icon color is not very prominent. Recommended to change the color or add a black border to make it stand out more. Additionally, in Figure 8, it would be better to zoom in, as the pink triangle is hard to find now.

**Response:** Thank you for the comment. We have modified Figure 7 & 8.

[Figure]

Figure 7 Comparison of satellite imagery and classification results from 2012 and 2021. Panels (a) and (c) show high-resolution satellite images for 2012 and 2021, respectively. Colored boxes in these images indicate the enlarged areas shown in (b) and (d). Panels (b) and (d) present the classification results of the corresponding enlarged regions using a U-Net model trained with 61 selected features. Triangles mark field survey locations.

[Figure]

Figure 8. The spatial distribution of ATE area changes from 2012 to 2021. ATE expansion is marked in dark cyan, reduction is marked in dark red, persistence is marked in dark blue, and field survey point in purple.

L364, similarly, provide the full name of "ATE" here.

**Response:** Thank you for the comment. We have revised the manuscript.

Responses to Reviewer #4

L33, Add reference

**Response:** Thank you for the comment. We revised the manuscript and added the reference as follows (L33):

"The island contains more than 200 mountains exceeding 3,000 meters in elevation (Kuo et al., 2022)."

L43-44, Only wildfires? Thre are other natural disturbances potentially influencing such an issue. You should also mention the relevance eof cascading effets between disturbances in this context.

**Response:** Thank you for the comment. We revised the manuscript and added the reference as follows (L41-45):

"However, these shifts are also influenced by other drivers, including land-use history, altered disturbance regimes (e.g., fire, landslide, windthrows), herbivory pressure, and species-specific physiological traits. Moreover, cascading effects among these disturbances can further amplify ecological responses and accelerate treeline dynamics (Wang et al., 2016; Johnson et al., 2017; Du et al., 2018; Mohapatra et al., 2019, Stritih et al., 2024; Lu et al., 2025)."

L45-60, This section should be moved to the Discussion, where is generally useful to compare the study with previous researches, specifically focusing on the novelty of your study respect to the available literature.

**Response (45-73):** Thank you for the comment. Following the reviewer's advice, we have moved the paragraph summarizing previous treeline remote sensing studies from the Introduction to the Discussion and expanded it into a new subsection titled "4.1 Comparison with previous alpine treeline ecotone remote sensing studies." This section now highlights the novelty of our study, emphasizing the integration of ultra-high-resolution WorldView-2 imagery and U-Net for detecting krummholz and forest transitions in a subtropical alpine environment. We revised the manuscript as follows (L307-323):

"Recent advancements in remote sensing technology have enabled extensive studies on alpine treelines using imagery at various spatial resolutions (Garbarino et al., 2023). For example, Xu et al. (2020) employed Landsat (30 m) data to assess treeline–climate relationships in China, reporting an upward shift of ~50 m per 1°C increase in temperature. At medium to high resolution, Rösch et al. (2022) achieved over 90% classification accuracy for Pinus mugo in the Alps using PlanetScope (3 m) and Sentinel-2 (10 m) data, emphasizing the value of multi-source data fusion. At very high resolution, Terskaia et al. (2020) combined aerial orthophotos (1–2 m) and WorldView-2 imagery (0.5 m) to quantify shrub and tree encroachment in Alaska, detecting substantial vegetation transitions over six decades.

Building on prior work, fine-scale mapping of alpine treeline ecotones (ATEs) remains difficult because transitional vegetation is spatially heterogeneous, often includes stunted or shrubby forms such as krummholz, and exhibits subtle spectral/structural gradients at meter scales (e.g., Bader et al., 2021; Nguyen et al., 2022). Our study uses ultra-high-resolution WorldView-2 imagery (0.4 m) and machine learning workflows to detect fine-scale transitions within the ATE (~400 ha) in Taiwan. Concretely, we

show that integrating spectral bands, vegetation indices, and texture (GLCM) features at sub-meter resolution enables reliable separation of krummholz from closed-canopy forest—an underrepresented class distinction in many alpine studies (cf. Korznikov et al., 2021; Nguyen et al., 2022). This demonstrates the novelty and practical value of combining modern machine-learning segmentation with ultra-high-resolution imagery to fine-scale analyze the alpine treeline ecotone (ATE) in subtropical mountain environments. Related recent work similarly highlights the need for meter-scale approaches to capture ATE patterns and dynamics (Zou et al., 2022; Carrieri et al., 2024)."

L61-73, as above. Here, it should be better to simply adress the relevance of R, SVM models, as well as U-Net models in this context, of course with references to same recent papers, therefore moving to the aims of your research and respective innovative aspect.

**Response:** Thank you for the comment. We revised the paragraph to better highlight the study's aims and innovations as follows (L46-67):

"Machine learning is increasingly being combined with high-resolution remote sensing to enhance land-cover and forest-type classification. Among the numerous algorithms, each model has its own strengths. Random forests (RF) and support vector machines (SVM) have gained widespread use due to their robustness and effectiveness in processing multispectral data with limited training samples (Belgiu and Drăguţ, 2016; Jombo et al., 2020; Jackson and Adam, 2021). RF, in particular, exhibits strong interpretability and stability in heterogeneous environments. In contrast, deep learning models such as U-Net demonstrate superior ability to capture both spectral and spatial information, achieving high segmentation accuracy in complex landscapes (Ronneberger et al., 2015; Freudenberg et al., 2019; Wagner et al., 2019). Recent comparative studies further demonstrate that RF and SVM remain reliable and interpretable choices for multispectral classification when training data is limited or imbalanced. At the same time, U-Net and other convolutional neural network (CNN) architectures generally provide superior spatial accuracy and boundary delineation in high-resolution or well-labeled datasets. Furthermore, transferability analysis shows that U-Net models generally have better generalization capabilities in large or heterogeneous regions, while RFs tend to perform more consistently in small sample or sparsely labeled scenarios (Boston et al., 2022; Ge et al., 2021; Nigar et al., 2024).

In Taiwan, many alpine forest studies have been conducted through field surveys using an ecological approach at relatively small spatial scales, focusing on flowering phenology and growth assessment for specific tree species (Chiu et al., 2022; Liao et al., 2023a; Kudo et al., 2024). In recent years, Chung et al. (2021) used Landsat 8 imagery combined with support vector machine (SVM) classification to examine timberline dynamics on Taiwan's highest peak, Yushan, revealing the influence of temperature on timberline shifts. The Xue Mountain, Taiwan's second-highest peak, has also been the subject of long-term ecological monitoring (Chung et al., 2021; Liao et al., 2023b). However, extensive targeting alpine treeline ecotone (ATE) dynamics remains lacking. This study provides the first comprehensive analysis of changes in the ATE landscape in Taiwan's Xue Mountain glacial cirque region. It uses ultra-high-resolution WorldView-2 satellite imagery with Random Forest (RF) and U-Net models. The aim is to quantify spatiotemporal changes between 2012 and 2021."

L88, As done with ATE, explain the term with respective reference.

**Response:** In this study, krummholz, representing stunted, shrub-like trees typically found at high elevations near the treeline and shaped by wind or snow pressure (Liao et al., 2023a). The definition of krummholz was provided in (L78-79 and L191). Relevant explanation and details can be found in Liao et al., 2023a.

L91-95, Fig. 1: the boundaries of the study area are not clearly shown in the Figure. Is not neither clear the variation in elevations from the DTM.

Please, enlarge the RGB map, show boundaries of the study area, remove the DTM simply defining min and max elevation in the next, and clearly indicate the number of pictures with sequential numbers / letters (a,b,c...1,2,3...), also updating the text and the caption in accordance.

**Response:** Thank you for the comment. We have modified Figure 1 and revised the manuscript.

[Figure]

Figure 1. Study area. (a) Geographic location of Shei-Pa National Park in north-central Taiwan. (b) Treeline ecotone study area located in the Xue Mountain glacial cirques within Shei-Pa National Park. (c) WorldView-2 image showing the research area with topographic contours.

L122, Restate better.

**Response:** Thank you for the comment. The sentence has been revised for clarity as follows (L114):

"Both images were captured in the autumn season when vegetation had entered its dormant phase."

L123, GNSS is the more correct term.

**Response:** Thank you for the correction. The term "GPS" has been replaced with "Global Navigation Satellite System (GNSS)" throughout the manuscript to ensure technical accuracy. (L115)

L126, Remove the column, not useful.

**Response:** Thank you for the comment. The "Data quantization (Bits)" column in Table 1 has been removed as suggested.

L128-129, Add reference.

**Response:** Thank you for the comment. We revised the manuscript and added the reference as follows (L125-126):

"The reflectance spectrum of plant leaves can reflect their internal physiological status, such as chlorophyll content, water content, intercellular spaces, and cell walls (Croft et al., 2014; Xu et al., 2023; Neuwirthová et al., 2024; Špundová et al., 2024)."

L136, not use the future, but restate like "in this study, 11 vegetation indices was used". Please, update it in accordance alongside the entire manuscript.

**Response:** Thank you for the comment. We revised the manuscript as follows (L134):

"In this study, 11 vegetation indices were used, as summarized in Table 2."

L141-144, I suggest to restate these sentences in a better way. It is not clear wahat is the point: relevance to extract ground object (why?), or relevance of texture (?)...please explain better.

**Response:** Thank you for the comment. We revised the manuscript as follows (138-141):

"With improvements in satellite image resolution, a single ground object may consist of multiple pixels, making spatial information increasingly important for image interpretation (Wang et al., 2015). Texture features describe the spatial arrangement and structural patterns of objects within an image, providing complementary information to spectral reflectance. This allows for better discrimination of land cover types with similar spectral characteristics."

L146-151, Move to Discussion. Here, simply state your methodology based on previosu researches, but the comparison with other works as suggegsted before should be moved to the final part (Discussion) of the manuscript (of course, underlining the novelty of your research in this regard).

**Response:** Thank you for the suggestion. The purpose of citing Guo et al. (2020) and Sibiya et al. (2021) was to reference their parameter settings rather than to perform a comparative analysis. To clarify this, the paragraph in the Methods section has been revised to emphasize the methodological reference only, as follows (143-146):

"Following the parameter settings suggested by previous studies (Guo et al., 2020; Sibiya et al., 2021), texture features were extracted to enhance spatial information for classification. In this study, a moving window size of $7 \times 7$ was applied based on their findings, which provided an effective balance between detail and noise in texture analysis."

L163, Add reference.

**Response:** Thank you for the comment. We revised the manuscript and added the reference as follows (L157):

"The process began by randomly sampling the data to create training datasets. After each sampling, the selected data points were returned to the dataset for the next round of sampling (bootstrap sampling) (Breiman, 2001)."

L168, was.

L168, Better explain for readers.

**Response (L168):** Thank you for the comment. We revised the manuscript as follows (L162-164):

"The final classification result was determined by aggregating the predictions of all decision trees through a majority voting approach, which means that each tree casts one "vote" for a class label, and the class receiving the most votes becomes the final prediction."

L169, Add reference.

**Response:** Thank you for the comment. We revised the manuscript and added the reference as follows (L165):

"To evaluate the importance of each feature, the Random Forest model uses the Gini Index (Breiman, 2001)"

L169, What "node" stands for should be explained, or alternatively add a reference.

**Response:** Thank you for the comments. We revised the manuscript as follows (L165-166):

"which measures the impurity of a node. A node represents a point in the tree where the dataset is split based on a feature, with each node divided using the best split among a random subset of explanatory variables (Breiman, 2001)."

L173, Please explain what "feature importance" stands for in this context.

**Response:** Thank you for the comments. We revised the manuscript as follows (L171-172):

"Feature importance quantifies how much each variable reduces node impurity and contributes to improving classification accuracy across all trees in the forest (Belgiu and Drăguţ, 2016; Breiman, 2001; Chen et al., 2023)."

L178, Restate better, since is not clear how this sentence is linked (logically) with the previosu and the next ones.

**Response:** Thank you for the comments. We revised the manuscript as follows (L174-175):

"Ronneberger et al. (2015) proposed the original U-Net model, which was devolved from the fully convolutional network (FCN) and was initially designed for biomedical image segmentation."

L179, Add reference.

**Response:** Thank you for the comment. We revised the manuscript and added the reference as follows (L176):

"The U-Net model consists of a contracting path (downsampling) and an expanding path (upsampling) (Ronneberger et al., 2015)."

L202, Add reference to support this decision.

**Response:** Thank you for the comment. We revised the manuscript and added the reference as follows (L198):

> "The dataset split was performed at the patch level, to avoid spatial autocorrelation and data leakage (Roberts et al., 2017)."

L202, Parameters are not explained.

**Response:** Thank you for your comments. The parameters used in the Kappa coefficient (Po and Pe) were already described in detail in Lines 213–214, where their definitions and corresponding formulas are presented. Since this information is already included, no additional changes were required.

L222, features.

L225-226, So, how such a misclassification influenced your results? You should discuss how to resolve such an issue, and show data supporting the fact that this didn't influenced obtained outcomes and models performances, since misclassification and errors between classes in both segmentation and next classification steps is a crucial point to be taken into account in similar works.

To support the proposed metodology and overall your results, you need to better consider such aspects, specifically in this case when you state that effective and evident misclassification occurred in your data-elaboration process.

L228-229, You need to show results supporting your statement.

**Response (L222, 225-226, 228-229):** Thank you for the valuable comment. We have revised Section 3.1 to discuss the potential influence of misclassification and to demonstrate, through performance metrics and field validation, that such errors did not significantly affect results or model performance. We revised in Lines 225-248, Section 3.1, and Discussion Lines 338-345, Section 4.2, as follows:

> "This study employed Random Forest (RF) and U-Net models with four feature combinations to examine land cover classes in Taiwan's Xue Mountain glacial cirques in the alpine treeline ecotone (ATE) region. Four land cover classes —bare land, forest, krummholz, and shadow —were investigated using feature combinations of spectral bands (8 features), vegetation indices (13 features), and texture features (56 features). The classification results of the RF and U-Net models with four feature combinations were compared in detail (Fig. 5 and Table 5). In general, the RF model demonstrated stable, robust classification performance across various feature dimensions. Specifically, the average F1-score of the RF model ranged from 0.823 to 0.839, the overall accuracy (OA) ranged from 0.817 to 0.830, and the Kappa coefficient ranged from 0.751 to 0.768 (Table 5). Among all classes, shadow and bare land achieved the highest F1-scores, both exceeding 0.85, while forest and krummholz maintained moderate but stable accuracy, ranging from 0.75 to 0.83. Additionally, the combination 4 yielded the highest F-1 score in forest and krummholz classes, indicating that the RF model improved when vegetation indices and texture features were combined with spectral information.

Furthermore, the U-Net model exhibited a marked improvement after incorporating more features. The F1-score for the forest class increased significantly from 0.609 for feature combination 1 (spectral bands only) to 0.828 for combination 4 (spectral, vegetation indices, and texture features). Likewise, the F1-score for krummholz improved from 0.696 to 0.778. Bare land and shadow also maintained high accuracy above 0.82 across all combinations. The U-Net's overall performance metrics (F1-score of 0.840, OA of 0.838, and Kappa of 0.778 in combination 4) surpassed those of RF, indicating that the U-Net model benefited substantially from integrating spectral, vegetation, and texture information.

Overall, the results showed that incorporating vegetation indices and texture features improved classification performance, particularly for vegetation classes in the U-Net model. Based on the higher F1-score in combination 2 than combination 3, it implied that vegetation indices contributed more than texture features. However, the highest F1-score was obtained in combination 4, indicating a complementary effect from vegetation indices and texture features. Additionally, the consistency between the classified ATE and field-observed forest–krummholz transitions further confirmed the classification's reliability. Overall, both models maintained stable performance across different feature combinations, supporting the robustness of the proposed approach."

"Despite the overall satisfactory classification performance, some confusion between forest and krummholz was observed due to their similar canopy structures and spectral reflectance. This misclassification occurred mainly along the transition between dense forest to stunted krummholz. However, this issue had only a limited influence on the overall outcomes. Field survey validation confirmed that the classified treeline boundaries were consistent with the observed forest–krummholz transitions in situ, and both RF and U-Net models maintained high accuracies (OA > 0.83, Kappa > 0.76). Therefore, the local confusion slightly affected boundary precision but did not alter the overall trend of the alpine treeline ecotone. To further minimize this effect in future work, incorporating structural features, such as LiDAR-derived canopy height models, could improve discrimination between forest and krummholz and enhance classification reliability."

L250, I think that time-consuming analysis is out of scope here. I suggest to remove this, simply highligjting the relevance to find out th best methodoogy able to reduce the time necessary for data elaboration. Infact, you should also consider the computation capabilities of different work stations, for example, as well as the possibility to improve performances (in terms of time-consuming) of data processing with various strategies (specifically concerning modeling parameters, programming languages, etc...)

**Response:** The time-consuming analysis (previous Section 3.2) has been shortened and integrated into Section 3.1 to improve conciseness of manuscript (L256-261).

"Based on the RF and U-Net model results, a further feature importance analysis was conducted to assess individual features in combination 4, comprising 77 features, including spectral bands, vegetation indices, and texture features. The feature importance analysis results revealed that the cumulative contribution achieved 95% interpretability with 61 features. Additionally, the OA and Kappa values improved slightly to 0.842 and 0.784, respectively. Moreover, computation time was reduced by 14.3% due to fewer features (Table 6). According to the feature ranking results, spectral bands and vegetation

indices ranked higher than texture features, with SEVI, Y, B, G, and NDVI2 identified as the top five features (Fig. 6)."

L254, GNSS. Update in all the manuscript.

**Response:** The term "GPS" has been replaced with "Global Navigation Satellite System (GNSS)" throughout the manuscript to ensure technical accuracy.

L273, Increase the size of X-axis label. Label of the Y-ais is missing. Increase also the size of the title and numbers along the Y-axis as well.

**Response:** The figure was redrawn to improve readability. (L266 Figure6)

[Figure]

Figure 6. Feature importance ranking derived from the Random Forest model. Features are ranked based on their contribution to classification accuracy, with the top-ranked features including SEVI, Y (yellow), B (blue), G (green), and NDVI2. Most of the top features are spectral bands and vegetation indices, while texture features rank lower.

L276-277, See comment at line 250.

**Response:** We moved Table 6 to match the updated manuscript in L262. The time-consuming analysis (previous Section 3.2) has been shortened and integrated into Section 3.1 to improve conciseness of manuscript (L256-261).

L280, The figure is not clear for many reasons:
- what the black boxes stand for is not stated, neither in the text
- If results are similar using 77 and 61 features, it should be enought to simply tell it in the text, maybe the figure is not so necessary ? Please, justify your choice to include Fig.6, in accordance with the aims of your research.

**Response:** We have removed Figure 6.

L285, This was not stated before, but only here. You have to better explain the scope of ground truth validation data at the beginning of the manuscript, not at the end.
In addition, you say that trees' positions were collected along an elevation gradient, but no more details are explained in this regard (e.g., what elevation gradient? why you proceed in this way?...) these aspect need to be addressed at the beginning of the paper.

**Response:** To avoid misunderstanding, we rephrased the paragraph and explained the field observation in sections 2.3 and 3.2.

L288-289, Of course, but you could try to reduce the presence of shadow in data elaboration process, to improve classification results. This is not considered in the text: I suggest to better explain, in the methodology explanation, how to potentially reduce the presence of shadow adjusting models parameters, and the relevance of such an aspect.

See also comment for table 7.

**Response:** Thank you for your comments. In the methodology section 2.3, we have explained that we applied histogram matching technique to ensure the radiometric consistency across the two images (L113-115).

L293, No mention of this in the methodology section. Please edit.

**Response:** Thank you for your comments. We have added it to section 2.6.5. as follows (L216-222):

> "The bootstrap resampling method was a nonparametric approach used to estimate the variability and confidence intervals (CIs) of a statistic by repeatedly resampling with replacement from the original dataset. It enabled robust inference without assuming a specific data distribution (Efron and Tibshirani, 1993). The percentile method was commonly used, in which the 2.5th and 97.5th percentiles of the bootstrap distribution defined the 95% CI (Davison and Hinkley, 1997). To ensure stable and reliable estimates, between 1000 and 10,000 bootstrap iterations were generally recommended (Davison and Hinkley, 1997), with at least 5000 replicates providing sufficient accuracy for most applications (Carpenter and Bithell, 2000)."

L299-302, Results as shown here are not persuasive and acceptable: the tree line detection by classification outcomes is not clear in your figure: I suggest to enlarge the size of the image, add a basemap on the background, adjust transparency and remove the visualization of the shadow class, to better visualize tree line detection from classification procedure. Here it just seems a mosaic of classes with no clear evidence of tree line detection, despite the presence of ground truth point (in addition: add simply dots for ground points).

In addition: looking at RGB images, treeline is not so evident. Please choose a more clear area where tree line is evident from RGB images and therefore by the classification outcomes, considering the comment above.

**Response:** Thank you for the comment. We have modified Figure 7.

[Figure]

Figure 7 Comparison of satellite imagery and classification results from 2012 and 2021. Panels (a) and (c) show high-resolution satellite images for 2012 and 2021, respectively. Colored boxes in these images indicate the enlarged areas shown in (b) and (d). Panels (b) and (d) present the classification results of the corresponding enlarged regions using a U-Net model trained with 61 selected features. Triangles mark field survey locations.

L303, The reasons of increasing presence of shadows are not stated, neither how to reduce their presence in the classification outcomes). Please consider this issue in the text, adding accurate discussion of such aspects looking at the aims of your research.

See also comment at line 288

L304, This should depend on classification accuracy and data preparation. Consider this issue about shadows looking at the previous comments.

**Response:** Thank you for your comments. The presence of shadow may reveal evidence of ecological changes, such as growing trees or a denser canopy. Additionally, the increase in shadow also aligns well with our findings of forest expansion. Furthermore, terrain-related shadows could provide important information about the microenvironment for further analysis of habitat conditions.

L307-308, Some improvement are needed:

- remove the DTM on the background, is not useful and create confusion.

- instead, add a basemap with transparency adjustements

- field investigation points are not visible

- this is not the entire study area, or yes? it is not clear

- change colors with a good palette of omogeneous colors

- increase the size of the figure, moving legend and items on the left side

- reduction class is not visible (?)

**Response:** Thank you for the comment. We have modified Figure 8.

[Figure]

Figure 8. The spatial distribution of ATE area changes from 2012 to 2021. ATE expansion is marked in dark cyan, reduction is marked in dark red, persistence is marked in dark blue, and field survey point in purple.

L317, Update including sections highlighted in the comments above at the beginning of the manuscript.

**Response:** Thank you for the comment. We carefully edited the manuscript with update subtitle for each section.

L337, Too colloquial, update in a more formal way.

**Response:** Thank you for the comments. We revised the manuscript as follows (L352):

> "In this study, a total of 77 features were derived from the satellite imagery, including 8 spectral bands, 13 vegetation indices, and 56 texture features."

L342, Adjust in accordance.

**Response:** Thank you for the comments. We revised the manuscript. (L355-357)

> "Notably, most of the top-ranked features were spectral or vegetation index variables, whereas texture features contributed less to the classification. The feature selection slightly improved the overall accuracy (+0.4%) and the Kappa coefficient."

L343-344, Use indirect syntax. As above.

**Response:** Thank you for the comments. We revised the manuscript as follows (L357-360):

> "Although OA was used as the primary selection criterion, the analysis also confirmed that the selected features maintained or improved F1-scores for the forest class, the primary focus of detecting treeline changes. It should be noted that optimizing overall accuracy (OA) values may sometimes overlook minority or ecologically important classes."

L363, Conclusions need to be improved: in-depth discussion of the usefulness of your research is missing, as well as concerning aspecs including ecology, changes in habitats and species presence, influence of ATE shift in natural disturbances occurrence and social risks, social impacts of ATE changes over time, changes in forest management practices, climate change and future scenarios...
Please consider all these aspects (and other) and improve the Conclusions of your research.

**Response:** We reorganized the conclusion section to address ecology aspects related to our findings. We emphasized our contribution and innovation of demonstrating the potential of high-resolution satellite imagery for long-term ecological monitoring.

L362, Use a more accurate term.

**Response:** Thank you for the comments. We rephrased the sentence as follows. (L385)

> "This study investigates changes in the alpine treeline ecotone (ATE) of the Xue Mountain glacial cirques in Taiwan from 2012 to 2021, utilizing WorldView-2 imagery integrated with Random Forest and U-Net models."

The reviewer marked several sentences for deletion directly in the annotated PDF.

**Response:** We sincerely thank the reviewer for the detailed annotations and constructive suggestions. Most of the sentences marked for deletion have been removed as suggested. In addition, several related sentences in the same sections were also revised to improve clarity, readability, and consistency throughout the manuscript.